# Post-translational regulation of retinal IMPDH1 in vivo to adjust GTP synthesis to illumination conditions

Anna Plana-Bonamaisó[1,2], Santiago López-Begines[1], David Fernández-Justel[3], Alexandra Junza[4,5], Ariadna Soler-Tapia[1], Jordi Andilla[6], Pablo Loza-Alvarez[6], Jose Luis Rosa[1,7], Esther Miralles[8], Isidre Casals[8], Oscar Yanes[4,5], Pedro de la Villa[9,10], Ruben M Buey[3], Ana Méndez[1,2,7]*

[1]Department of Physiological Sciences, School of Medicine, Campus Universitari de Bellvitge, University of Barcelona, Barcelona, Spain; [2]Institut de Neurociències, Campus Universitari de Bellvitge, University of Barcelona, Barcelona, Spain; [3]Metabolic Engineering Group, Department of Microbiology and Genetics. University of Salamanca, Salamanca, Spain; [4]CIBER of Diabetes and Associated Metabolic Diseases (CIBERDEM), Madrid, Spain; [5]Metabolomics Platform, IISPV, Department of Electronic Engineering, Universitat Rovira i Virgili, Tarragona, Spain; [6]ICFO-Institut de Ciencies Fotoniques, The Barcelona Institute of Science and Technology, Castelldefels, Spain; [7]Institut d'Investigació Biomèdica de Bellvitge (IDIBELL), Campus Universitari de Bellvitge, University of Barcelona, Barcelona, Spain; [8]Centres Cientifics i Tecnològics (CCiTUB), University of Barcelona, Parc Científic de Barcelona, Barcelona, Spain; [9]Physiology Unit, Dept of Systems Biology, School of Medicine, University of Alcalá, Madrid, Spain; [10]Visual Neurophysiology Group-IRYCIS, Madrid, Spain

*For correspondence:
mendezzu@idibell.cat

Competing interests: The authors declare that no competing interests exist.

**Abstract** We report the in vivo regulation of Inosine-5´-monophosphate dehydrogenase 1 (IMPDH1) in the retina. IMPDH1 catalyzes the rate-limiting step in the de novo synthesis of guanine nucleotides, impacting the cellular pools of GMP, GDP and GTP. Guanine nucleotide homeostasis is central to photoreceptor cells, where cGMP is the signal transducing molecule in the light response. Mutations in IMPDH1 lead to inherited blindness. We unveil a light-dependent phosphorylation of retinal IMPDH1 at Thr$^{159}$/Ser$^{160}$ in the Bateman domain that desensitizes the enzyme to allosteric inhibition by GDP/GTP. When exposed to bright light, living mice increase the rate of GTP and ATP synthesis in their retinas; concomitant with IMPDH1 aggregate formation at the outer segment layer. Inhibiting IMPDH activity in living mice delays rod mass recovery. We unveil a novel mechanism of regulation of IMPDH1 in vivo, important for understanding GTP homeostasis in the retina and the pathogenesis of adRP10 IMPDH1 mutations.

## Introduction

Mutations in inosine monophosphate dehydrogenase 1 (IMPDH1), the enzyme responsible for the first and rate-limiting step in GTP synthesis, are associated to severe forms of inherited blindness. At least nine mutations have been associated to the RP10 form of autosomal dominant retinitis pigmentosa, that primarily manifests as night blindness and gradually progresses to loss of central vision: R224P (*Kennan et al., 2002*); D226N (*Bowne et al., 2002*); R231P (*Grover et al., 2004*); T116M, V268I, G324D, H372P (*Bowne et al., 2006a*); K238E and K238R (*Wada et al., 2005*). Together they account for about 1% of adRP cases (*Sullivan et al., 2013*). *IMPDH1* mutations have also been

associated to rare autosomal dominant Lebers Congenital Amaurosis (adLCA), characterized by nearly complete blindness from an early age: R105W and N198K (*Bowne et al., 2006a*). Despite the ubiquitous nature of guanine nucleotide synthesis, clinical manifestations of *IMPDH1* mutations are limited to the retina, for reasons that are not yet understood.

RP10 mutations are allegedly 'gain-of-function' mutations, given that IMPDH1 knock-out mice present only a mild retinopathy (*Aherne et al., 2004*). Mutations do not directly affect IMPDH1 catalytic activity in vitro (*Aherne et al., 2004*; *Mortimer and Hedstrom, 2005*; *Xu et al., 2008*). IMPDH1 capacity to bind single-stranded nucleic acids (*Mortimer et al., 2008*; *Hedstrom, 2008*) as well as IMPDH1 tendency to aggregate (*Aherne et al., 2004*; *Tam et al., 2008*) have been proposed to contribute to the pathophysiology. More recently, structural studies have provided a new framework to interpret the effect of mutations, by revealing the allosteric inhibition of eukaryotic IMPDHs by GDP/GTP binding to the Bateman domain (*Buey et al., 2015*). GDP/GTP binding sites at the Bateman domain overlap with several of the residues mutated in adRP10 (*Buey et al., 2015*).

There are two IMPDH isozymes, IMPDH1 and IMPDH2, that are 84% identical. While most human tissues show only basal IMPDH1 and high IMPDH2 expression, the retina is one of a few exceptions where IMPDH1 predominates (*Hedstrom, 2009*). In the retina, major unique IMPDH1 spliced forms outweigh the canonical 514aa protein (*Bowne et al., 2006b*).

Purine nucleotide homeostasis is vital for many basic functions of the cell. Cells use different pathways to synthesize purine nucleotides and balance the guanine and adenine pools (*Figure 1*). In de novo biosynthesis, a purine ring is assembled over ribose 5′-phosphate by sequential enzymatic steps that use precursors of the carbohydrate and amino acid metabolism (*Zhao et al., 2013*; *Pedley and Benkovic, 2017*). In this pathway, inosine monophosphate (IMP) is the first molecule with a complete purine ring and is the common precursor for GTP and ATP synthesis (*Figure 1*). IMPDH catalyzes the first committed step to GTP synthesis. In de salvage pathways, nucleotides are produced from recycled purine bases (*Figure 1*).

IMPDH monomers consist of a catalytic and a regulatory domain. The catalytic domain is a $(\beta/\alpha)_8$ TIM barrel that catalyzes two sequential reactions that convert IMP to xanthosine monophosphate (XMP) in a NAD dependent manner (*Hedstrom, 2009*). GMP acts as a competitive inhibitor of enzymatic activity at this level, by binding to the IMP pocket (*Hedstrom, 2009*). The regulatory Bateman domain consists of two cystathionine β-synthase (CBS) repeats that act as ATP and GTP sensors to mediate allosteric modulation of catalytic activity (*Buey et al., 2015*; *Labesse et al., 2013*; *Anthony et al., 2017*; *Buey et al., 2017*; *Fernández-Justel et al., 2019*). In eukaryotes, binding of adenine nucleotides (ATP/ADP/AMP) induces the formation of extended octamers that remain fully

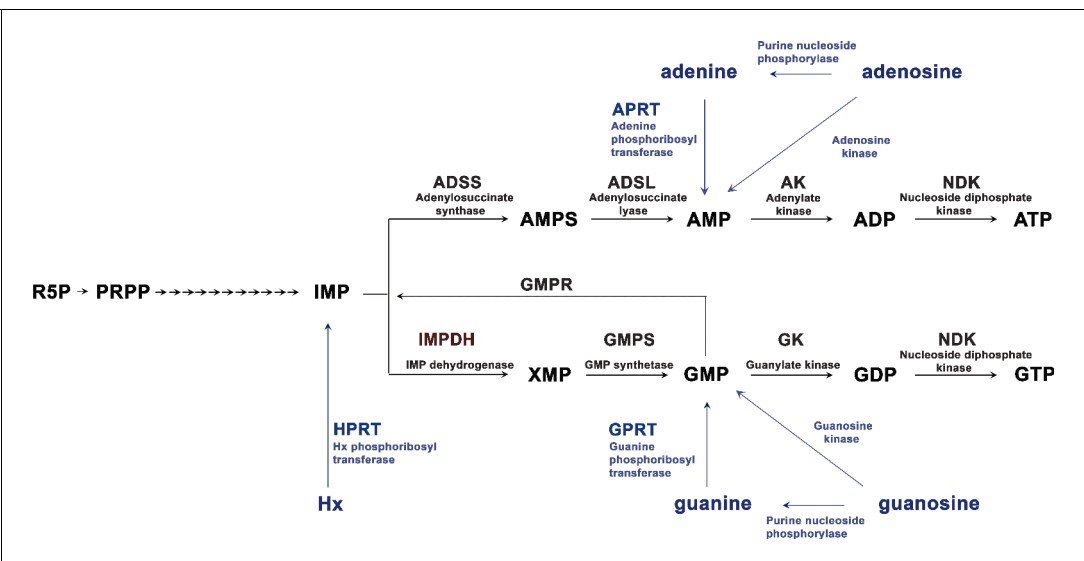

**Figure 1.** De novo and *salvage* pathways of purine synthesis in eukaryotic cells. Metabolites and enzymes of the de novo pathway in black, and the salvage pathway in blue. R5P, ribose 5 -phosphate; PRPP, phosphoribosylpyrophosphate. The arrows from PRPP to IMP represent sequential enzymatic activities of the purinosome complex. IMPDH1 is responsible for the rate-limiting and first committed step in de novo GTP biosynthesis.

active, while GDP/GTP binding induces the formation of compacted octamers with significantly reduced catalytic activity (*Buey et al., 2015*; *Labesse et al., 2013*; *Buey et al., 2017*).

The fact that most *IMPDH1* mutations associated to RP10 overlap with GDP/GTP binding sites at the CBS domain and result in diminished GDP/GTP negative allosteric control of human IMPDH1 in vitro (*Buey et al., 2015*; *Fernández-Justel et al., 2019*) points to the physiological relevance of this allosteric mechanism in vivo.

IMPDH can form mesoscale filamentous structures in mammalian cells (spicules or cytoophidia) under conditions that require higher IMPDH activity to keep with GTP demand (*Liu, 2010*; *Chang et al., 2015*; *Aughey and Liu, 2016*; *Keppeke et al., 2018*), like high rates of proliferation, or Gln/Ser/folate-metabolite deficiency (*Chang et al., 2015*; *Keppeke et al., 2018*; *Calise et al., 2016*). Formation of these reversible fiber-like subcellular structures is believed to transiently boost IMPDH activity (*Chang et al., 2015*; *Keppeke et al., 2018*). Reversible formation of IMPDH fibers could be recently induced in vitro from recombinant proteins, revealing that most adRP10 IMPDH1 mutants form filaments but fail to disassemble them due to their inability to sense adenine and guanine nucleotides at the Bateman domain (*Fernández-Justel et al., 2019*).

Little is known about the relevance of IMPDH1 catalytic activity in photoreceptor cells of the retina or its physiological regulation in vivo. We here report the first posttranslational modifications of IMPDH1 in native retinal tissue, associated to dark/light states. We show that the predominant retinal splice form of IMPDH1 is phosphorylated at the Bateman domain (T159/S160) in response to light in vivo; and that this phosphorylation desensitizes the enzyme to GDP/GTP allosteric inhibition of catalytic activity in vitro. Together, these results reveal a novel light-dependent regulatory mechanism that impedes IMPDH allosteric inhibition, facilitating the elevated GTP levels required for phototransduction. Supporting the proposed mechanism, we show that exposure of living mice to bright light results in an increase in the global flux through de novo purine nucleotide synthesis in the retina, which correlates with a progressive accumulation of IMPDH1 aggregates at the outer segment layer. Furthermore, inhibition of IMPDH activity in living mice by intravitreal injection of IMPDH inhibitors has a moderate but consistent effect on mass rod response recovery measured by electroretinogram. Thereby, our results point to the de novo GTP synthesis being relevant to sustain the GTP pool during constant light exposure, likely to withstand the increased cGMP turn-over. This study gains insight into the complex in vivo regulation of IMPDH1 to maintain GTP homeostasis with changing illumination, central to rod dark/light physiology and relevant to understand *IMPDH1*-inherited retinal dystrophies.

## Results

### Dark/light-dependent phosphorylation of retinal IMPDH1

Phosphoproteomic analysis of dark- and light-adapted bovine retinas (see Methods) revealed three phosphorylation sites in IMPDH1 (*Figure 2A–B*). One phosphorylation event was detected at the Bateman domain, assigned to Thr[159] or Ser[160] with a 50:50 probability (numbering corresponding to canonical bovine IMPDH1β of 514aa with Uniprot code A0JNA3). This numbering correlates with the canonical human IMPDH1β. The peak areas of the precursor ions for the monophosphorylated peptide 154–169 in 3 dark and 3 light biological replicates revealed a light preference of the observed phosphorylation event (light/dark log2 fold change = 1.67 with p=0.03). The light/dark log2 fold change of phosphopeptides representative of well characterized light- or dark-dependent phosphorylation events in rhodopsin (*Wilden, 1995*; *Lee et al., 2002*; *Mendez et al., 2000*), phosducin (*Lee et al., 1990*; *Lee et al., 2004*) and GRK1 (*Lee et al., 1982*; *Palczewski et al., 1992*) is shown in *Figure 2—figure supplement 1* as a quality control of the phosphoproteomic analysis.

Residues Thr[159] and Ser[160] map within the CBS motif 1 of the Bateman domain, in close proximity to Asn[198], Arg[224] and Asp[226] mutated in adLCA or adRP10 (*Figure 2C–D*), and are directly involved in the binding of purine nucleotides at the allosteric nucleotide binding site 1 (*Fernández-Justel et al., 2019*).

Phosphorylation at residue Ser[416] was also identified, but observed to a similar extent in dark and light conditions (light/dark log2 fold change = 0.33 with p=0.08). This residue maps at the mobile flap in the catalytic domain (residues 412–432, in green in *Figure 2E*), required in the conformational transition step that activates the hydrolysis reaction during catalysis (*Hedstrom, 2009*).

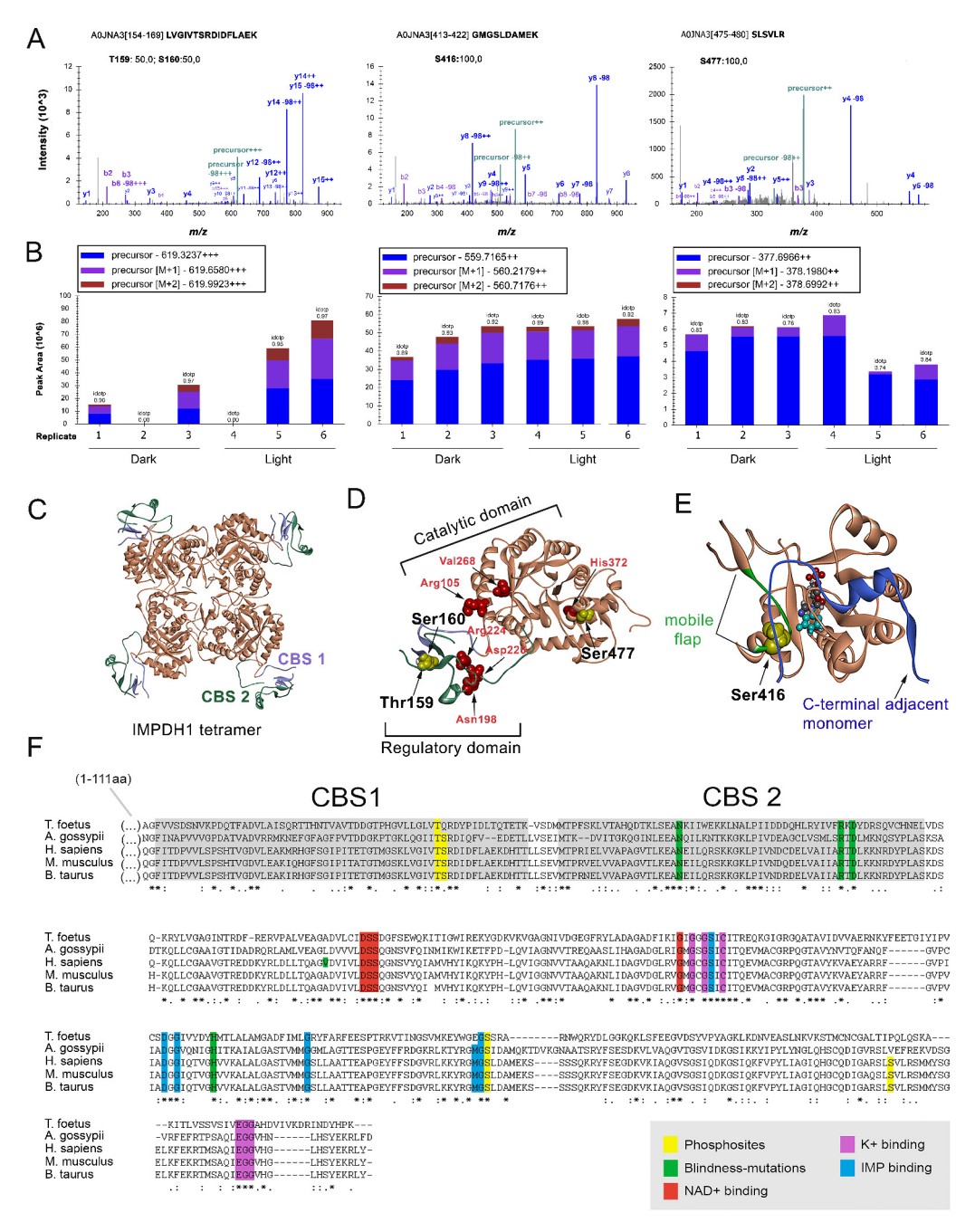

**Figure 2.** IMPDH1 phosphorylation sites identified in a phosphoproteomic study of dark/light-adapted bovine retinas. (**A**) Mass spectra of identified IMPDH1 phosphorylated peptides. Residue numbers refer to the canonical bovine isoform β (514aa) [Uniprot A0JNA3]. For peptide 154–169 only one phosphorylation was detected, that could not be unequivocally assigned to T159 or S160. Phosphorylation at peptides 413–422 and 475–480 was at S416 and S477, respectively. (**B**) Peak areas of the precursor ions corresponding to the identified phosphopeptides of IMPDH1. Samples 1–3 are dark-adapted retinas; 4–6 light-exposed retinas (biological replicates). T159/S160 are preferentially phosphorylated in light, S477 in dark, and S416 indistinctly. (**C**) Ribbon diagram of IMPDH in its tetrameric conformation [pdb: 1jcn], showing the TIM barrel catalytic domain in coral, and the regulatory Bateman domain with two copies of cystathione beta-synthase (CBS) sequence in slate blue and green. (**D**) Monomer conformation [pdb: 1jcn] showing T159 and S160 at CBS1 in the Bateman domain; and S477 at the COOH-terminus. Disease associated mutations, depicted in red, are proximal to T159/S160 (Asn198; Arg224; Asp226) or to S477 (His372). (**E**) The catalytic domain of AgIMPDH, with the C319 loop in coral, the COOH-terminus of an adjacent monomer in blue and the mobile flap in green. IMP depicted in space fill model [pdb:4xfi]. (**F**) Alignment of IMPDH from T. foetus (prokaryotic), A. gossypii (filamentous fungus), *H. sapiens*, *M. musculus* and *B. taurus* IMPDH1 canonical isoforms. Phosphosites highlighted in yellow.

*Figure 2 continued*

The online version of this article includes the following figure supplement(s) for figure 2:

**Figure supplement 1.** Dark/light fold-change of peak areas from control phosphopeptides from phosducin, GRK1 and rhodopsin shown as a quality control of the phosphoproteomic analysis.

Finally, phosphorylation of IMPDH1 at $Ser^{477}$ occurred preferentially in the dark-adapted state (light/dark log2 fold-change = −0.50 with p=0,19). $Ser^{477}$ maps to a region that has not been involved in the catalytic process. Remarkably, all the phosphorylated residues are conserved in mammalian IMPDHs (*Figure 2F*).

To further characterize IMPDH1 phosphorylation in vivo we performed an in situ metabolic labeling assay. Retinas from dark-adapted mice were dissected and incubated in Locke´s buffer with $^{32}P$-inorganic phosphate for 90 m in the dark, to allow for $^{32}P$ incorporation in the ATP pool. Retinas were then kept in the dark or exposed to 2000 lux light for 5 m. IMPDH1 was immunoprecipitated from supernatant and membrane fractions with an anti-IMPDH1 pAb generated against the canonical bovine IMPDH1β of 514aa (see Materials and methods). This antibody immunoprecipitated all IMPDH1 isoforms in the murine retina (immunoblots in *Figure 3A–B*). $^{32}P$-incorporation was limited to the largest and most abundant isoform in murine retinas, the 603/604aa retinal-specific spliced form (*Bowne et al., 2006b*) (autoradiographs in *Figure 3A–B*). Phosphorylation of 603/604aa IMPDH1 occurred to a high extent in dark- and light-retinas, both in the soluble and membrane fractions, *Figure 3C*.

To obtain insight into the predominant phosphorylated species in dark and light, immunoprecipitated IMPDH1 from $^{32}P$-labeled retinas was resolved by isoelectrofocusing (IEF), *Figure 3D*. Three abundant species were detected in dark and light conditions: unphosphorylated IMPDH1-603/604aa, IMPDH1-603/604aa-1P, and IMPDH1-603/604aa-2P.

The same species IMPDH1-0P, −1P and −2P were observed by IEF separation of retinal homogenates from living mice that were either dark-adapted or exposed to 2000 lux light for 5, 20 and 60 m. Monophosphorylated forms are the predominant species in both dark and light conditions (*Figure 3E*). Please note that the intensity of the di-phosphorylated band in the IEF radiograph results from the incorporation of two $^{32}P$ atoms. Therefore, a similar intensity of the mono- and di-phosphorylated IMPDH1 bands actually reflects that the di-phosphorylated form is half as abundant as the mono-phosphorylated form.

Measured isoelectric points (IP) of IMPDH1-603/604aa-0P, −1P and −2P correlated well with IP predicted values, *Figure 3E*.

Taken together, our results show that the most abundant isoform of IMPDH1 in the retina, which is a retina-specific spliced form, is phosphorylated to a high extent in vivo. Nearly two thirds of the protein are phosphorylated in vivo, with mono- and di-phosphorylated forms being present but monophosphorylated species predominating under dark or physiological light conditions.

## In vitro effects of phosphorylation on IMPDH1 catalytic activity

While IMPDH1-603/604aa is the prevalent isoform in murine retinas, two isoforms predominate in human retinas: IMPDH1α (546aa) and IMPDH1γ (595aa) (*Bowne et al., 2006b*). In order to study the effect of phosphorylation on the catalytic activity of IMPDH1, the recombinant human IMPDH1α (546aa) and its individual phosphomimetic mutants were expressed in bacteria and purified to homogeneity. The S160D-hIMPDH1α (546aa) and S477D-hIMPDH1α (546aa) mutants showed very similar reaction kinetics and Michaelis-Menten parameters as the wildtype protein (*Figure 4—figure supplement 1*), indicating that phosphorylation at these sites was unlikely to affect the $K_m$ or $V_{max}$ of the enzyme. However, the S416D-hIMPDH1α (546aa) mutant showed a slower reaction (*Figure 4—figure supplement 1*). This result was not surprising, given that S416 maps at the mobile flap that determines the open or closed conformation of the enzyme during enzyme catalysis (*Hedstrom, 2009*).

We confirmed the effect of $Ser^{416}$ phosphorylation by measuring the effect on kinetics and IMP dependence of the S416D substitution in the canonical and in both prevalent hIMPDH1 retinal isoforms: hIMPDH1α (546aa) and hIMPDH1γ (595aa) (*Figure 4A–B*). *Figure 4C* shows that S416D substitution in the mobile flap had a significant effect at inhibiting the enzyme, that was higher for the

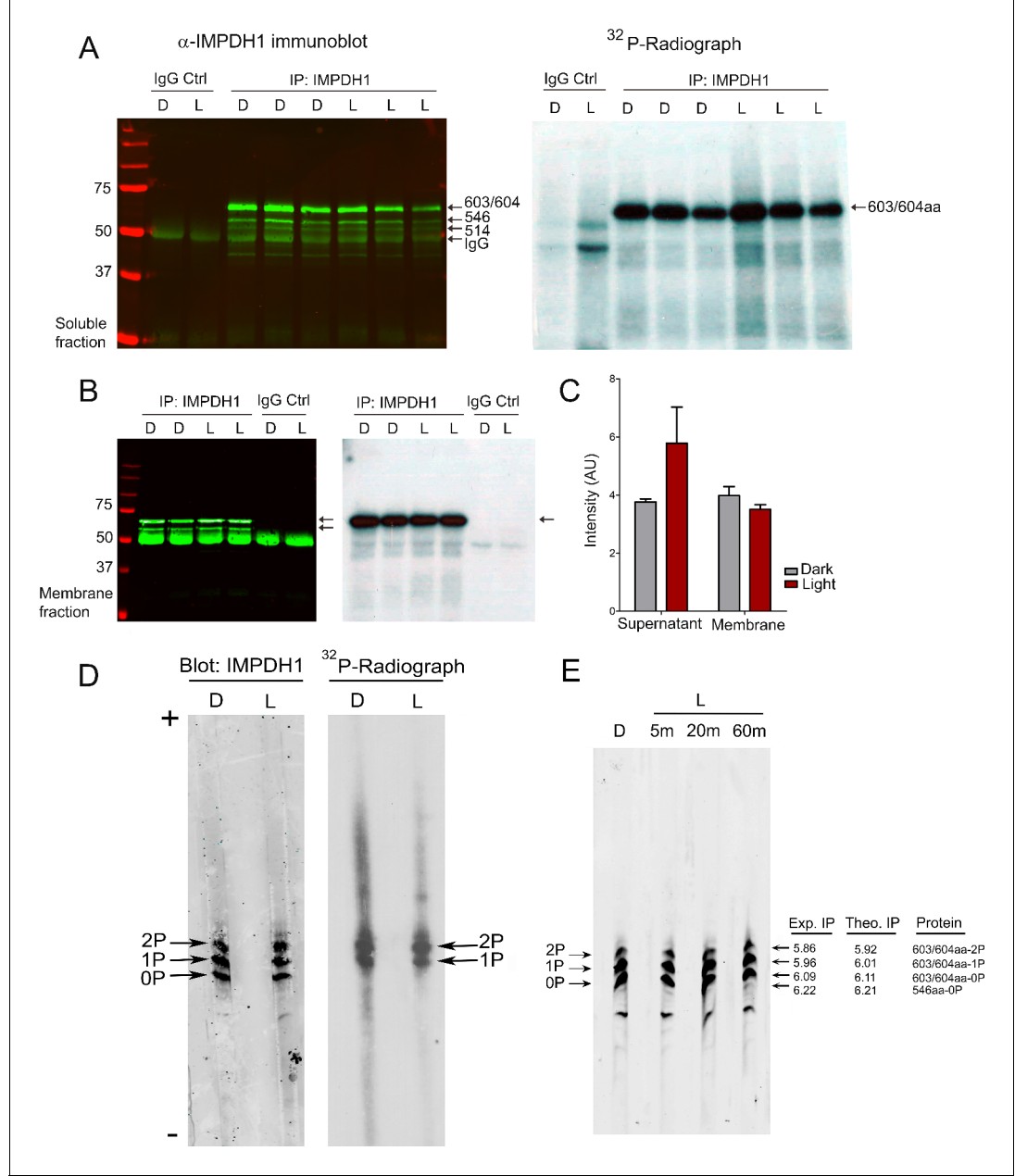

**Figure 3.** The predominant retinal spliced variant of IMPDH1 is phosphorylated to a high extent in intact retinas. (**A**) Immunoblot of IMPDH1 immunoprecipitated from soluble fractions of $^{32}$P-labeled retinas, and corresponding autoradiograph. In situ metabolic labeling was performed in dim red light, with retinas from dark-adapted mice incubated in Locke's with $^{32}$P$_i$ for 90 min at 37°C, and then kept in the dark or exposed to light of 2000 lux for 5 min. IMPDH1 was immunoprecipitated in 3 biological replicates. $^{32}$P$_i$ incorporation was observed at mIMPDH1-603/604aa isoform, in both dark and light conditions (autoradiograph). (**B**) Immunoblot and autoradiograph of IMPDH1 immunoprecipitated from membrane fractions. $^{32}$P$_i$ incorporation was observed at IMPDH1-603/604aa, at dark and light conditions (2 biological replicates). (**C**) Average intensity of $^{32}$P-labeled 603/604aa bands normalized by immunoblot signal, for dark and light samples. IMPDH1 is substantially phosphorylated under dark and light, both in soluble and membrane fractions. A tendency to a higher phosphorylation in light in soluble fractions (p=0.180) and in dark in membrane fractions (p=0.308) was apparent, with no statistical significance. (**D**) Isoelectrofocusing (IEF) of immunoprecipitated IMPDH1 samples from $^{32}$P$_i$-labeled retinas revealed the presence of mono- and di-phosphorylated forms of IMPDH1-603/604aa in dark and light. Observed bands were: mIMPDH1-603/604-0P; mIMPDH1-603/604-1P and mIMPDH1-603/604-2P. Please note that the di-phosphorylated form incorporates 2 radiolabeled $^{32}$P atoms per protein molecule, and therefore the band intensity of the di-phosphorylated form in the autoradiograph reflects twice its abundance. (**E**) IEF of retinal homogenates (soluble fractions) from 16 hr dark-adapted mice or mice that were exposed to 5, 20 or 60 min of 2000 lux light after pupil dilation. Measured isoelectric points (IP) of bands correlated well with theoretical IP for the mIMPDH1 isoforms indicated.

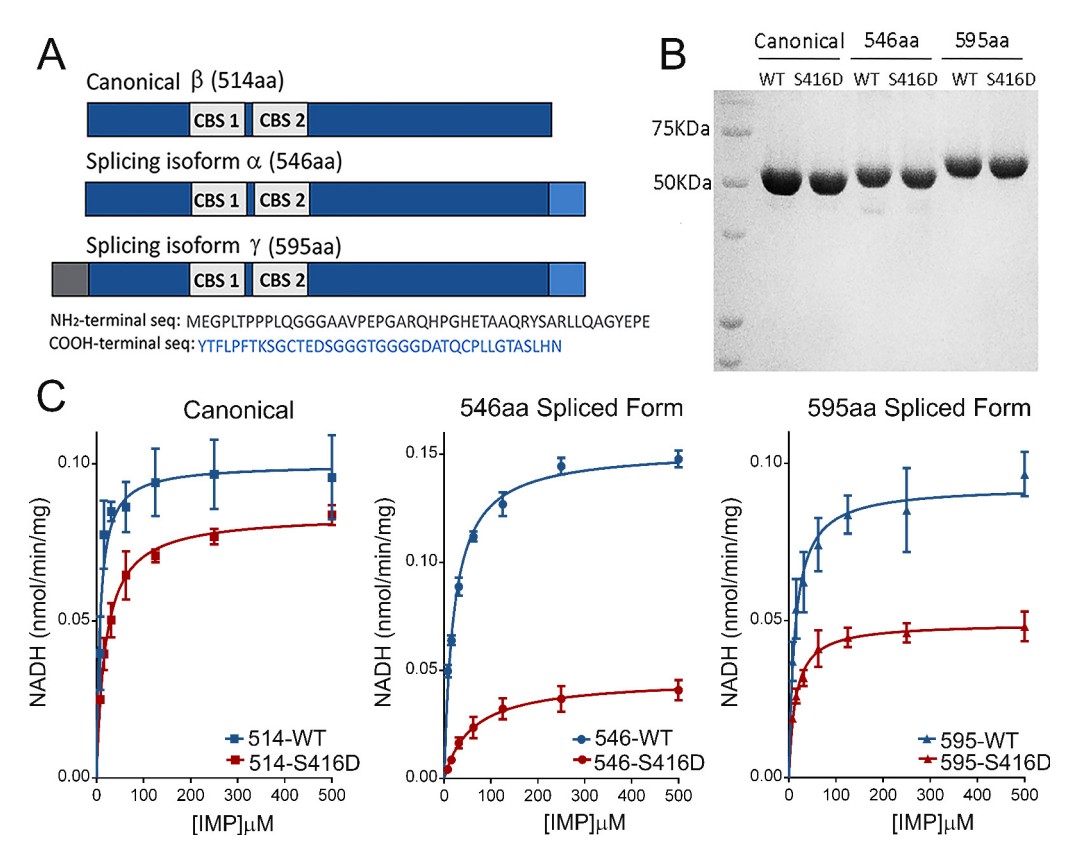

**Figure 4.** Effect of S416D phosphomimetic substitution on enzymatic activity. (**A**) Main splice variants of hIMPDH1: hIMPDH1α (546aa) and hIMPDH1γ (595aa) contain extended sequences at the COOH- terminus (546aa) or both the NH$_2$- and COOH-termini (595aa). (**B**) S416D mutant recombinant proteins were rigorously normalized to their wildtype counterparts preceding enzymatic analysis. (**C**) Effect of S416D substitution on the Michaelis-Menten kinetics of hIMPDH1β (514aa); hIMPDH1α (546aa); and hIMPDH1γ (595aa) isoforms. Mutation S416D increased the $K_m$ for IMP and decreased the $V_{max}$ significantly in hIMPDHα (546aa) and γ (595aa) splice variants, with the effect being maximal in the hIMPDH1α (546aa) isoform. Kinetic parameters were: hIMPDH1β (514aa) [$V_{max}$ = 0.099 ± 0.004, $K_m$ = 7.62 ± 1.69]; S416D/hIMPDH1β (514aa) [$V_{max}$ = 0.084 ± 0.002, $K_m$ = 19.13 ± 1.75]; hIMPDH1α (546aa) [$V_{max}$ = 0.152 ± 0.002, $K_m$ = 20.72 ± 1.32]; S416D/hIMPDH1α (546aa) [$V_{max}$ = 0.046 ± 0.002, $K_m$ = 59.59 ± 9.74]; hIMPDH1γ (595aa) [$V_{max}$ = 0.093 ± 0.003, $K_m$ = 12.86 ± 2.18]; S416D/hIMPDH1γ (595aa) [$V_{max}$ = 0.049 ± 0.001, $K_m$ = 13.95 ± 1.71]. Results represent the media and S.E.M of three independent experiments with three technical replicates each.

The online version of this article includes the following figure supplement(s) for figure 4:

**Figure supplement 1.** Effect of phosphomimetic substitutions on enzymatic activity.

retinal spliced forms of 546 and 595aa than for the canonical form. However, because phosphorylation at this residue was similar in dark and light conditions, we believe that the decrease in IMPDH1 catalytic activity associated to Ser$^{416}$ phosphorylation might occur in response to other signals such as nutritional stress but would have an effect unrelated to dark/light physiological conditions (see Discussion).

It has recently been established that GDP and GTP allosterically inhibit the catalytic activity of eukaryotic IMPDH enzymes in vitro (*Buey et al., 2015*; *Anthony et al., 2017*; *Buey et al., 2017*; *Fernández-Justel et al., 2019*). Thereby, we then tested whether phosphorylation could affect allosteric inhibition in vitro.

*Figure 5A* plots the normalized $V_{max}$ values ($V_{max}^{app}$ in the presence of 2 mM GTP or GDP divided by $V_{max}$ in the absence of nucleotide) for all the hIMPDH1β (514aa) phosphomimetic mutants as well as for the wild-type IMPDH1 enzyme. As expected, phosphorylation at Thr$^{159}$/Ser$^{160}$ had an obvious effect on the allosteric inhibition of IMPDH1 in vitro (*Figure 5A*). We then assayed the catalytic activity of the T159E/D mutants at different concentrations of GDP (*Figure 5B*) and GTP

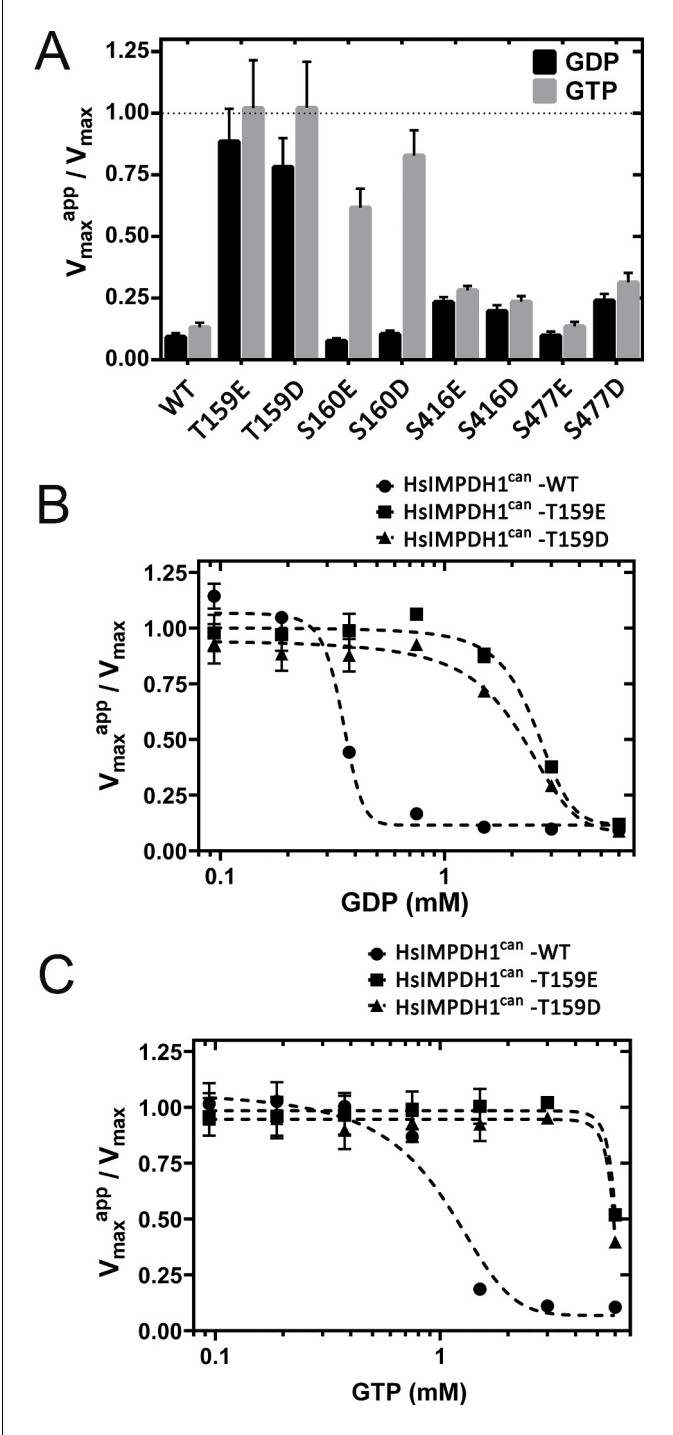

**Figure 5.** Effect of phosphomimetic mutations on GDP and GTP allosteric regulation of IMPDH1 activity.
(A) $V_{max}^{app}/V_{max}$ for the indicated phosphomimetics mutants. (B) $V_{max}^{app}/V_{max}$ for the T159E and T159D mutants, as a function of GDP concentration. (C) $V_{max}^{app}/V_{max}$ for the T159E and T159D mutants, as a function of GTP concentration. Results shown are representative of two independent experiments.
The online version of this article includes the following figure supplement(s) for figure 5:

**Figure supplement 1.** HPLC determination of nucleotide levels in dark-adapted or light-exposed in situ retinas.

(*Figure 5C*), that yielded $K_{1/2}$ values for enzyme inhibition around 5-fold higher than for the wild-type IMPDH1 (*Table 1*).

Taken together our results show that phosphorylation at Thr[159]/Ser[160] desensitizes the enzyme to the allosteric inhibition exerted by GTP and GDP. Given the light-dependence of this phosphorylation event, these results indicate that IMPDH1 would be susceptible to GDP/GTP inhibition in darkness but desensitized to this allosteric regulation in light.

In support of GDP/GTP allosteric regulation of IMPDH1 having physiological relevance, is the fact that photoreceptor cells present much higher GTP levels than most cells (*Traut, 1994*; *Zhao et al., 2015*; *Sumita et al., 2016*; *Biernbaum and Bownds, 1979*; *Berger et al., 1980*; *Ostroy et al., 1990*).

We have assessed GTP levels (as a function of ATP levels) in dark/light retinas by high pressure liquid chromatography analysis using murine retinas obtained from dark-adapted mice, and kept in situ for 5 m in darkness or exposed to bright light (2000 lux light). It must be noted that although our nucleotide determinations were done in whole retinal extracts, guanine nucleotide levels largely reflect nucleotide content in photoreceptor cells, as cGMP and GTP levels are reduced to less than 20% their values in retinas that lack the photoreceptor cell layer (*rd1* mouse model *Du et al., 2016*). The levels of GMP, GTP, AMP and ATP in dark and light conditions are presented in *Figure 5—figure supplement 1*, together with a representative HPLC chromatogram. Nucleotide peaks were assigned based on their retention time and absorbance spectrum, attained from nucleotide standards. Our results showed that GTP levels are at least equimolar with ATP levels, both in darkness and under light conditions. Note similar GTP and ATP peaks in a representative chromatogram (*Figure 5—figure supplement 1*). Another observation was that both AMP and GMP levels increased with light exposure.

The fact that GTP is equimolar or higher than ATP in the retina further confirms that GTP levels are higher in photoreceptor cells than in most cell types, where the ratio of ATP:GTP is between 3:1 to 5:1 (*Traut, 1994*; *Zhao et al., 2015*; *Sumita et al., 2016*).

## Constant bright light exposure results in IMPDH1 aggregation at the outer segment layer, and in increased flux towards de novo GTP and ATP synthesis

It has been reported that mammalian IMPDHs can form mesoscale macromolecular assemblies in mammalian cells, denoted as cytoophidia, when an increase in IMPDH activity is required to keep with GTP demand (*Aughey and Liu, 2016*; *Keppeke et al., 2018*; *Chang et al., 2015*; *Liu, 2016*). Recently it has been proposed that IMPDH1 cytoophidia are more resistant to GDP/GTP-mediated allosteric inhibition, which indicates that they would allow a boost of GTP synthesis when required (*Fernández-Justel et al., 2019*; *Keppeke et al., 2018*; *Duong-Ly et al., 2018a*). IMPDH1 cytoophidia are not noticeable in the retina in mice reared in standard 12 hr dark/12 hr light cycles, although they can be induced and clearly detected if retinas are treated with mycophenolic acid (MPA), *Figure 6—figure supplement 1*.

We next intended to assess whether IMPDH1 cytoophidia assembly could be induced by physiological conditions that increased GTP demand, such as exposing living mice to bright light for increasing time periods (*Figure 6*). The premise was to activate cGMP phosphodiesterase maximally by bright light, so that cGMP synthesis increased accordingly during light adaptation, consuming GTP in the process. *Figure 6A* shows the characteristic immunolocalization of IMPDH1 in the retina.

**Table 1.** K1/2 values (mM) for GDP and GTP.

Enzyme kinetics data were fitted by non-linear regression to the Michaelis-Menten equation to derive $V_{max}$ and $K_M$ values. The $V_{max}$ values *versus* GTP/GDP concentration were then adjusted to a sigmoidal dose-response function using the GraphPad software package.

| | GDP | GTP |
|---|---|---|
| HsIMPDH1β (514aa)-WT | 0.45 ± 0.03 | 0.97 ± 0.1 |
| HsIMPDH1β (514aa)-T159D | 2.19 ± 0.2 | ≈ 6 |

$K_{1/2}$ values are given in mM units (Mean ± SD), and are representative of two independent experiments.

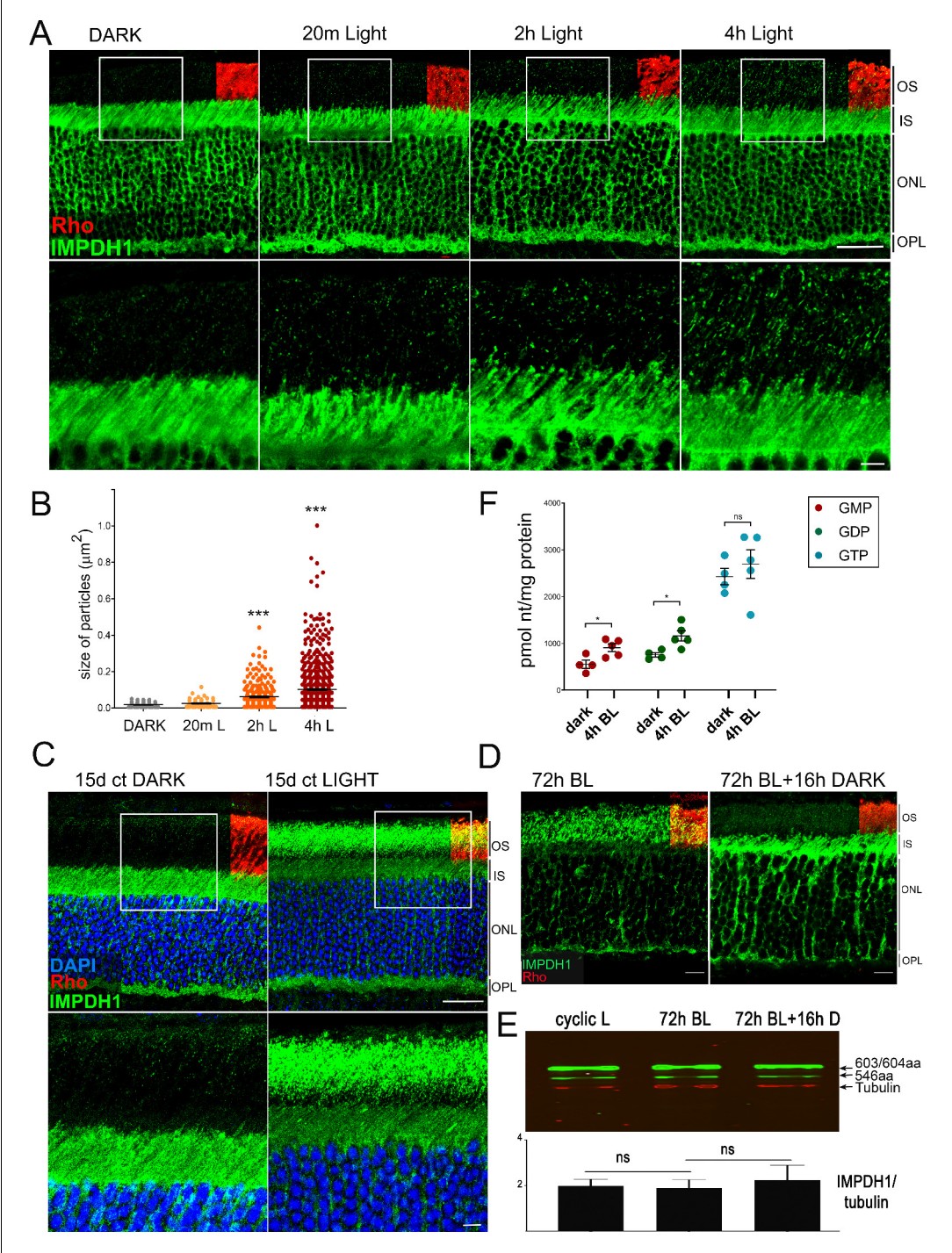

**Figure 6.** Accumulation of IMPDH1 aggregates at the outer segment layer in living mice exposed to constant bright light. (**A**) Murine retinal sections from dark-adapted mice or mice exposed to 1600lux-light for 20 m, 2 hr or 4 hr were immunostained for IMPDH1 (green) and co-immunostained for rhodopsin (red). Confocal images are average z-projections of 5 optical slices with 0.13μm-step size. Magnified frames from the outer segment layer show gradual accumulation of IMPDH1 aggregates with time of bright light exposure. (**B**) Size and number of IMPDH1 aggregates with time (Dark vs 2 hr light, p<0.0001; Dark versus 4 hr light, p<0.0001). Results are representative of two independent experiments. (**C**) Retinal sections from constant dark or 1600 lux constant light-reared mice for 15d, immunostained for IMPDH1 (green) and rhodopsin (red). Magnified inner/outer segment layers show a prominent accumulation of IMPDH1 signal at the outer segment layer, representing 30% of the total IMPDH1 signal in the retina. A basal IMPDH1 signal is already detected at the outer segment layer in dark-adapted mice. Results shown are representative of results obtained with two independent groups of mice. (**D**) IMPDH1 aggregate formation at the outer segment by 72 hr of bright light could be reversed by subsequent dark-adaptation (16 hr). Shown is a representative result observed in three independent groups of mice. (**E**) IMPDH1 expression in the retinas from these three groups of

*Figure 6 continued on next page*

*Figure 6 continued*

mice does not change, indicating that changes in IMPDH1 signal at the outer segment layer are due to protein translocation rather than to transcriptional regulation. (F) Retinal GMP, GDP and GTP levels (pmol/mg prot) determined by HPLC from retinal extracts from mice that had been dark-adapted or exposed to bright light (BL, 1600 lux) for 4 hr. GMP and GDP levels increased with light [p=0.021 for GMP and p=0.016 for GDP, with n = 4] whereas the GTP levels were maintained [p=0.802 for GTP, with n = 4]. OS: outer segment, IS: inner segment, ONL: outer nuclear layer, OPL: outer plexiform layer. Scale bar upper panels (20 µm) and bottom panels (5 µm).

The online version of this article includes the following figure supplement(s) for figure 6:

**Figure supplement 1.** Formation of IMPDH1 spicules in intact retinas in response to mycophenolic acid-treatment.

IMPDH1 signal is much stronger in photoreceptor cell layers than at inner layers of the retina. IMPDH1 signal is particularly intense at the inner segment, outer nuclear and outer plexiform layers, and stronger in rods than cones. Strikingly, exposure of living mice to 1600 lux white light after pupil dilation led to the gradual accumulation of IMPDH1 aggregates at the rod outer segment layer, that increased in number and size with time (*Figure 6A–B*). In contrast to MPA treatment that induced cytoophidia formation at the outer nuclear and inner segment layers (*Figure 6—figure supplement 1*), IMPDH1 aggregates induced by bright light accumulated at the rod outer segment layer. IMPDH1 accumulation was even more evident when mice were reared for 15d under constant light (1600 lux), *Figure 6C*. This light-dependent accumulation of IMPDH1 aggregates at the outer segment layer did not involve a change in the protein levels of IMPDH1, and could be reverted by subsequent dark-adaptation (*Figure 6D,E*). This result indicated that IMPDH1 translocates from the cell soma to the outer segment layer under bright light exposure, in a reversible manner.

To assess whether IMPDH1 catalytic activity increased in response to light in vivo, we determined the nucleotide levels in whole retinas from mice that had been dark-adapted or exposed to 4 hr of 1600 lux light. First, nucleotide determinations were done by HPLC. Subsequently, nucleotide determinations were done by LC coupled to tandem mass spectrometry (LC-MS/MS), following the intravitreal injection of a stable isotope form of labeled glycine. Because the amino acid glycine contributes carbon and nitrogen atoms to the scaffold on which the purine ring assembles, labeled glycine allowed us to analyze whether the incorporation of labeled Gly atoms into IMP, AMP and GMP increased with light in vivo.

The nucleotide determination by HPLC showed that GMP and GDP levels increased with 4 hr of light exposure, while GTP levels were maintained (*Figure 6F, ,Table 2*). No massive drop in the GTP levels was observed with bright light exposure, as reported in ex vivo retinas in other studies (see Discussion).

For the metabolic flux analysis in vivo, $^{13}C2;^{15}N$-Glycine was injected intravitreally in dark-adapted mice. Mice were then kept in the dark for 4 hr, or exposed to 1600 lux light for 4 hr. Whole retinal extracts were analyzed by mass spectrometry, to determine the levels of labeled and total cGMP, IMP, AMP, GMP, ATP and GTP, see Methods for experimental details.

**Table 2.** Nucleotide levels in dark- or light-adapted murine retinas, determined by HPLC.
Retinas were obtained from 16 dark-adapted mice that were either kept in the dark for 4 hr, or exposed to 4 hr bright light (1600 lux). HPLC determinations, with numbers indicating Mean ± S.E.M, with n = 4 biological replicates.

|  | Pmol/mg protein | |
|---|---|---|
|  | DARK | 4 hr BL |
| GMP | 557.21 ± 88.5 | 916.31 ± 81.1 |
| GDP | 762.57 ± 47.8 | 1174.11 ± 109 |
| GTP | 2461.53 ± 178.1 | 2737.29 ± 310.6 |
| AMP | 731.84 ± 125.4 | 1246.16 ± 69.8 |
| ATP | 1579.69 ± 110.5 | 2759.18 ± 226 |
| GTP/ATP | 1.56 | 0.99 |

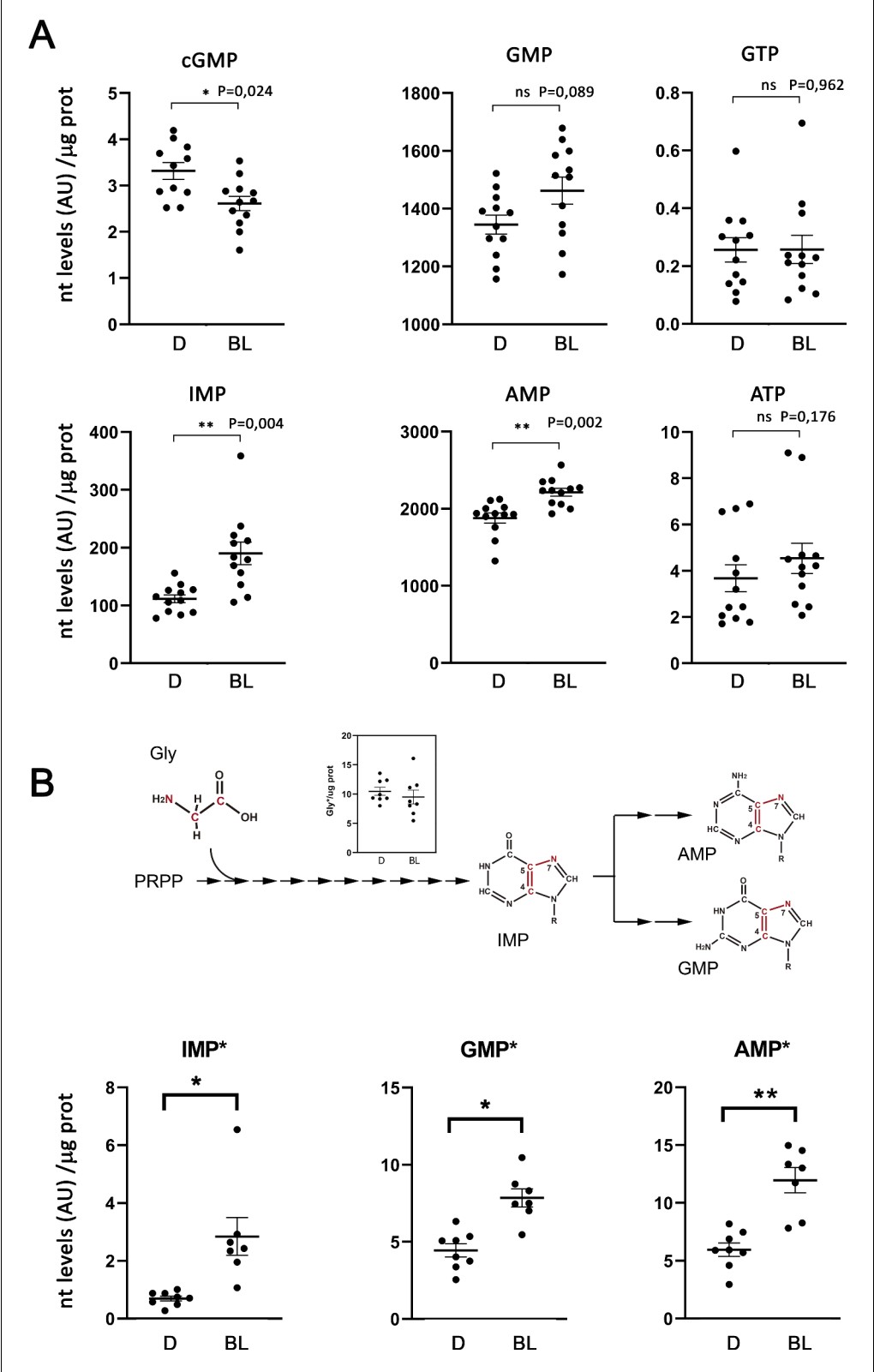

**Figure 7.** Light exposure increases the flux through de novo purine nucleotide synthesis in the retina. Metabolic flux analysis through de novo purine nucleotide synthesis was performed by injecting $^{13}C2;^{15}N$-Glycine intravitreally (to 4 mM final concentration in the vitreous) to dark-adapted mice, that were then kept in the dark or exposed to 1600 lux light for 4 hr. Total nucleotide levels and labeled nucleotide levels (that incorporated the labeled carbon and nitrogen atoms from labeled glycine) were determined by LC/MS-MS, and normalized per μg of protein in each sample. (**A**) Retinal levels of

*Figure 7 continued on next page*

*Figure 7 continued*

total cGMP, IMP, AMP, GMP, ATP and GTP in dark-adapted mice (D); or bright-light-exposed mice (BL). Nucleotide levels are expressed in arbitrary units (LC-MS/MS peak integration values) normalized per µg of protein in each sample. cGMP levels decreased with light exposure, as expected [p=0,024, n = 12]. IMP levels showed a clear increase with light exposure [p=0004]; that was accompanied by an increase in AMP and GMP levels [p=0002 for AMP; p=0089 for GMP]; whereas the ATP and GTP levels were maintained. (B) Retinal levels of labeled IMP, GMP and AMP, normalized per µg prot. The labeled atoms from Gly* that are incorporated into the purine ring scaffold are shown in the diagram. The inset on the diagram shows that the levels of Gly* measured at the retinas of injected eyes were similar in dark and bright light samples. Incorporation of Gly* into IMP, AMP and GMP increased under bright-light exposure, indicative of an increase in the overall flux of de novo purine nucleotide synthesis [IMP*, p=0,020; AMP*, p=0,003; GMP*, p=0,013, with n > 7]. In four eyes of the 'dark' group of mice, and five eyes of the 'bright light' group of mice the injection failed [the injected Gly* did not reach the retina, and Gly*~0 in retinal extracts] and therefore could not be taken into account for determination of labeled nucleotides.

*Figure 7A* presents the levels of total cGMP, IMP, AMP, GMP, ATP and GTP in dark and bright light conditions (expressed in arbitrary units of LC-MS/MS peak integration, normalized per µg of protein) for 12 biological replicas. As expected, mass spectrometry analysis detected the decrease in cGMP that follows adaptation to a light steady-state (*Barbehenn et al., 1986*), as well as the increase in GMP and AMP levels that we had observed in HPLC determinations. GTP and ATP levels were maintained. Mass spectrometry allowed the determination of IMP levels in dark and light, strikingly revealing a substantial increase in IMP levels in the bright light condition. IMP levels could not be determined by HPLC, as IMP is masked by the GMP peak under the gradient required to unequivocally separate GTP and ATP (*Sloan, 1984*).

*Figure 7B* shows the carbon and nitrogen atoms that glycine contributes to the purine ring, and presents the levels of labeled IMP, AMP and GMP in dark versus bright light conditions, normalized per µg of protein in each sample. Results are presented for 8 (dark) and 7 (light) individual retinas per group, because the intravitreal Gly* injection failed (Gly* did not reach the retina) in 4 (dark) or 5 (light) of the 12 injected eyes in each condition. Interestingly, IMP*, GMP* and AMP* increased in the bright light condition to the same extent as the total nucleotides, indicating that the increase in the total amount of IMP, AMP and GMP was due to an increase in de novo nucleotide synthesis. Gly was uptaken to similar levels in dark and light conditions in the injected eyes, as shown in *Figure 7B* inset.

Taken together, our results show that the global flux through the de novo synthesis of purine nucleotides increases with light. This increased flux could serve to maintain the ATP and GTP pools as their consumption increases in the phototransduction process.

## Inhibition of IMPDH catalytic activity delays mass rod recovery in electroretinogram responses

Two processes that increase GTP consumption upon light exposure in photoreceptor cells would be GTP hydrolysis by the GTPase transducin at the activation step, and GTP conversion to cGMP synthesis during recovery of the light response (*Mendez et al., 2001*; *Burns et al., 2002*).

To assess whether IMPDH1 catalytic activity was required to sustain the light response or rod mass response recovery, we recorded electroretinogram (ERG) responses simultaneously from both eyes of mice that were injected intravitreally with an IMPDH1 inhibitor (right eye) or control physiological saline buffer (left eye). The result of transiently inhibiting IMPDH activity on the rate of mass rod recovery after a saturating flash was tested using a paired-flash ERG paradigm (*Lyubarsky and Pugh, 1996*), and is shown in *Figure 8A–F*. Benzamide riboside (BZM) -converted to a dinucleotide in the cell after phosphorylation/adenylation- and mycophenolate mofetil (MMF) -an ester prodrug of mycophenolic acid (MPA)-, are both reversible selective noncompetitive IMPDH inhibitors. Inhibitors were injected intravitreously to an effective concentration of 80 µM (BZM); or 200 nM (MMF). Both drugs caused the characteristic SDS-PAGE mobility shift of the enzyme caused by drug binding (*Ji et al., 2006*), when retinas were obtained 20 m after intravitreal injection (*Figure 7G*), which demonstrated direct binding of the drugs to its target.

IMPDH1 inhibitors did not significantly affect the amplitude of the a-wave or b-wave of the test flash [3 Cd·s/m$^2$] in the high intensity range of mixed responses (*Figure 8E*), indicating no major effect on transducin activation. Following the paired flash paradigm, a test flash was triggered [3 Cd·s/m$^2$], and an identical probe flash was activated at 400, 600, 800, 1200, 1500 or 2000 ms

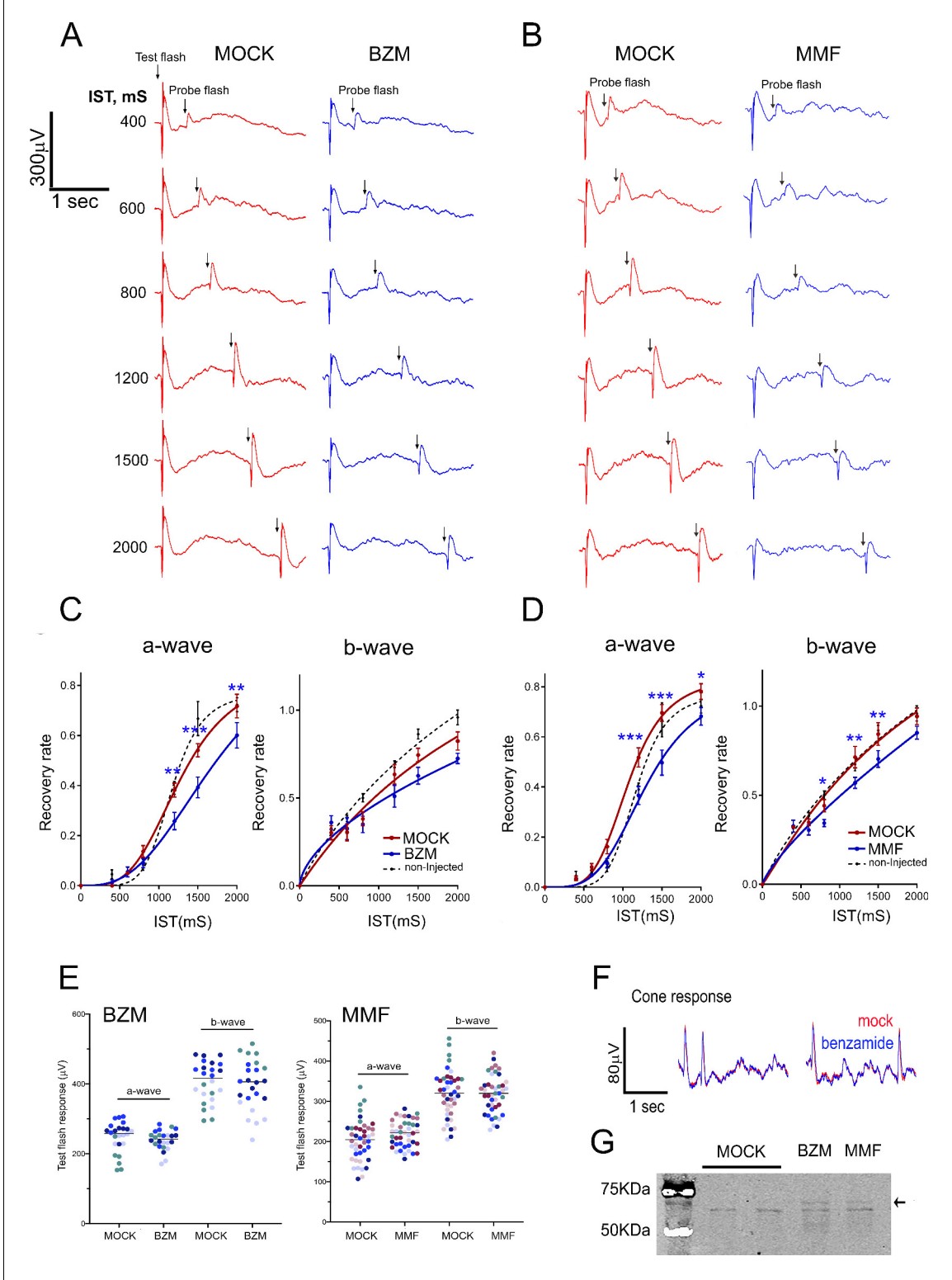

**Figure 8.** Effect of IMPDH inhibition on mass rod response recovery. (**A**) Simultaneous electroretinogram (ERG) recordings from both eyes of a mouse injected with benzamide (BZM, non-competitive IMPDH inhibitor) or drug dilution buffer (mock), in response to a probe flash triggered at increasing interstimulus times (IST) after a test flash. A slight but statistically significant delay in the recovery of the a-wave is observed at three interstimulus times. (**B**) A similar result is obtained when mycophenolate mofetil (MMF) -an ester prodrug of mycophenolic acid (MPA)- is injected instead of benzamide riboside. (**C**) Statistical analysis of mass rod response recovery (recovery rate plotted to interstimulus time, ERG paired flash paradigm) in benzamide-injected eyes versus corresponding control eyes. Overimposed black dashed lines are the curves obtained for control non-injected mice. Two-way

*Figure 8 continued on next page*

Figure 8 continued

ANOVA (mock vs drug injections and IST as factors) with uncorrected Fisher´s test for multiple comparisons, a-wave benzamide vs mock [1200 ms (p=0.0027); 1500 ms (p=0.0004); 2000 ms (p=0.0046), 4 mice analyzed]. (D) Statistical analysis for mycophenolate mofetyl (MMF)-injected mice: MMF vs mock [1200 ms (p=0.0002); 1500 ms (p<0.0001); 2000 ms (p=0.0132), 7 mice analyzed]. (E) Statistical comparison of a- and b-wave amplitudes of the test flash in the mock-injected left eye versus the drug-injected right eye in four BZM mice and seven MMF mice. Each mouse shown in a different colour. Mean a-wave and b-wave of the test flash (and SD) were: a-wave [MOCK: 246.8 ± 44.5 and BZM: 239.1 ± 28.2, n = 4]; and b-wave [MOCK: 405.0+58.5 and BZM: 395.0 ± 73.1]; and a-wave [MOCK: 203.0 ± 51.1 and MMF: 218.8 ± 33.4, n = 7]; and b-wave [MOCK: 318.3+58.9 and MMF: 318.2 ± 47.3]. (F) Superimposed pure-cone responses of the same mouse. (G) IMPDH1 mobility by SDS-PAGE is shifted by both benzamide and MMF in retinal extracts from intravitreally injected eyes.

intervals. Representative raw traces obtained for BZM and MMF are shown in *Figure 8A–B*. The percentage of recovery of a- and b-wave amplitudes was plotted to interstimulus time (IST), *Figure 8C–D*. Both BZM and MMF had a modest but statistically significant effect at delaying mass rod response recovery, while no alterations were observed in pure cone responses (*Figure 8F*).

Taken together our results indicate that IMPDH1 activity (de novo GTP synthesis) contributes to mass rod recovery by sustaining the GTP pool upon light exposure.

## Thr$^{159}$ and Ser$^{160}$ sites are phosphorylated by PKC in vitro

In order to identify the kinase/s responsible for the phosphorylation of the described residues of IMPDH1, the human IMPDH1α (546aa) isoform, and mutant forms with all of these 4 residues mutated to Gly (4KO, i.e. quadruple mutant T159**G**/S160**G**/S416**G**/S477**G**) or with individual phosphorylation sites restored in the 4KO (for instance, T159-only; T159**T**/S160**G**/S416**G**/S477**G**) were purified to perform in vitro phosphorylation assays with the kinases PKA, PKCα, PKG, CaCM-Kinase, CK II, PKB-Akt and AMPK. *Figure 9A* shows that PKCα phosphorylated the wild-type but not the 4KO enzyme, and phosphorylation was only restored in the mutants T159-only and S160-only, showing PKCα specificity for the Thr$^{159}$ and Ser$^{160}$ residues. No other kinase among the kinases tested showed specificity towards Thr$^{159}$/Ser$^{160}$; or towards Ser$^{416}$ or Ser$^{477}$ residues.

If PKC was the kinase responsible for IMPDH1 phosphorylation in vivo, then treating retinas in situ with a PKC inhibitor preceding/during light exposure should result in a decrease of the GTP pool. To test this, we treated dark-adapted in situ retinas with 50 nM bisindolylmaleimide for 20 m, and then kept the retinas in darkness or exposed them to bright light for 5 m. Nucleotide levels were determined by HPLC, *Figure 9B*. GTP levels were substantially decreased when retinas were exposed to light in the presence of the drug. This result supports that PKC is the kinase responsible for Thr$^{159}$/Ser$^{160}$ phosphorylation in vivo. Consistent with this, intravitreal injection of bisindolylmaleimide in living mice caused a delay in rod mass recovery similar to that caused by IMPDH inhibitors (*Figure 9E–F*).

Given the proximity of Thr$^{159}$ and Ser$^{160}$ to residues that are most prevalently mutated in adRP10 and rare LCA at the Bateman domain, we analyzed whether Thr$^{159}$/Ser$^{160}$ phosphorylation was affected in blindness associated mutations. For this, we purified the recombinant human IMPDH1α (546aa) isoform carrying individual mutations associated to RP10 (R224P, D226N, V268I and H372P) or LCA (R105W, N198K), and used them as substrates for phosphorylation reactions with PKCα in vitro (*Figure 9C*). Following a $^{32}$P-densitometry analysis, our data showed that the mutations N198K and R224P significantly reduced phosphorylation of IMPDH1 by PKCα, whereas H372P increased the phosphorylation level (*Figure 9D*).

Taken together our results point to light-dependent phosphorylation of IMPDH1 at the Bateman domain taking place in response to a light-triggered signaling pathway that involves PKC activation. Out of the few adRP10 mutations that do not affect GDP/GTP binding directly (*Fernández-Justel et al., 2019*), H372P enhanced PKC phosphorylation, which means it would cause GDP/GTP desensitization also. Taken together, our results stress the physiological relevance of GDP/GTP allosteric regulation of IMPDH1 in the maintenance of GTP homeostasis in photoreceptor cells of the retina, and highlights the disruption of this regulatory mechanism as causative of IMPDH1 associated blindness.

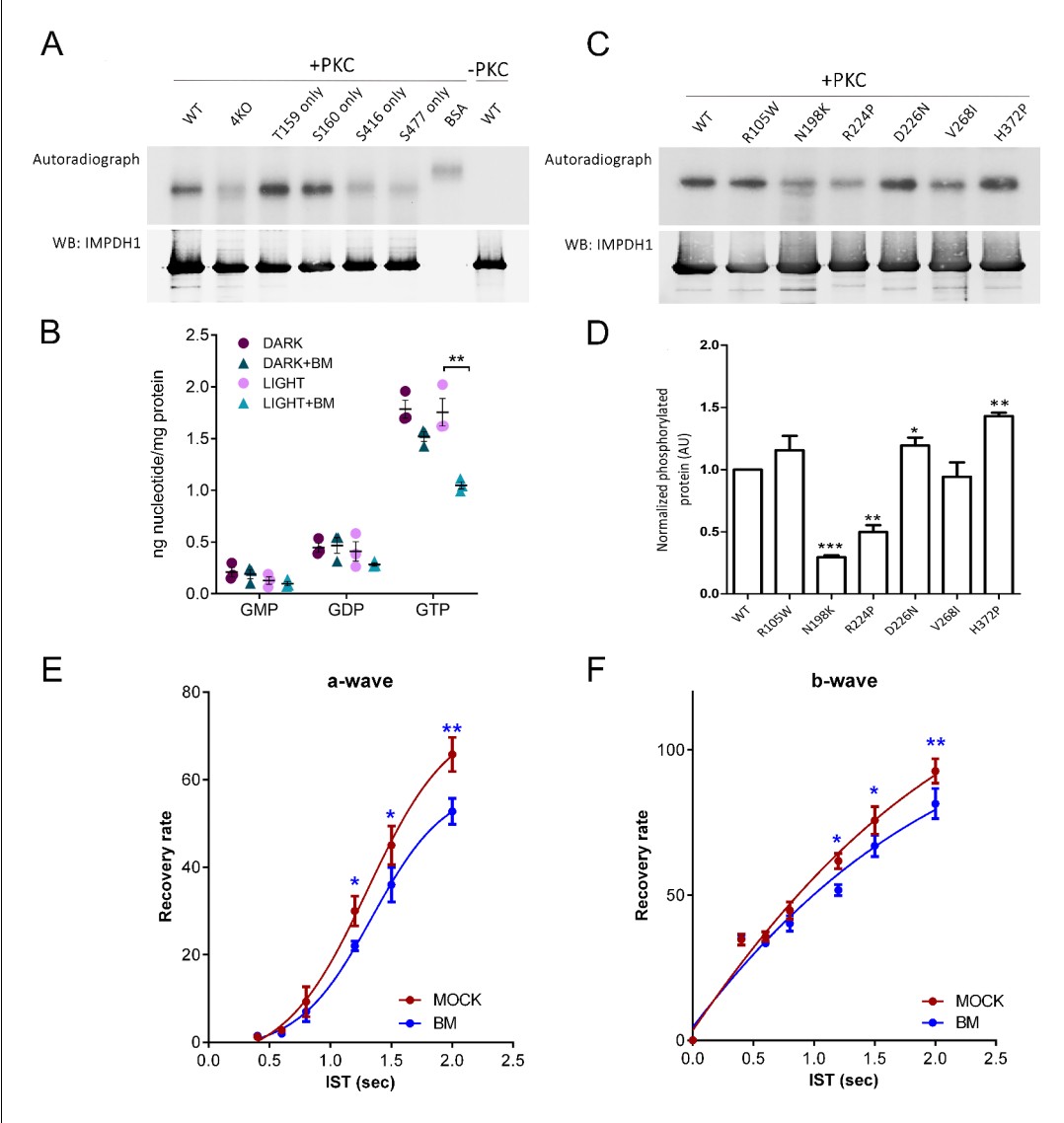

**Figure 9.** PKC phosphorylates T159 and S160 of hIMPDH1 in in vitro phosphorylation assays. (A) In vitro phosphorylation assays with PKCα kinase of hIMPDH1α (546aa); T159G/S160G/S416G/S477G-hIMPDH1α (546aa) (4KO); and forms of the protein with phosphorylation sites individually restored. 5 µg of purified protein were incubated with 4 µg/ml recombinant hPKCα in assay buffer with 0.05 mM PMA and 1 µl of $^{32}$P-γ-ATP (3000 Ci/mmol), for 30 min at 30˚C. Samples were resolved by SDS-PAGE, transferred to a nitrocellulose membrane, exposed to an X-ray film (autoradiograph), and immunoblotted for IMPDH1. PKC phosphorylated hIMPDH1α (546aa) selectively at T159 and S160. Results shown are representative of two independent experiments. (B) GMP, GDP and GTP nucleotide levels (nmol nt/mg of retinal protein) determined by ion-pairing high performance liquid chromatography (HPLC) in in situ dark-adapted or light-exposed retinas. Individual values and the Mean ± STDEV are indicated (3 biological replicates). In light-exposed retinas, GTP levels decreased substantially in the presence of 50 mM bisindolylmaleimide (unpaired t-test p=0.007); while other nucleotides did not show statistically significant changes between the dark and light conditions (n = 3). (C) In vitro PKC phosphorylation assay of hIMPDH1α (546aa) disease mutants R105W; N198K; R224P; D226N; V268I; H372P. (D) Average intensity of autoradiograph signal normalized by the immunoblot signal for blindness-associated mutations, expressed as a function of the wildtype levels. N198K and R224P mutations resulted in a decrease of phosphorylation [p=0.0001 and p=0.004 versus the wildtype, n = 3 independent experiments]; while H372P led to an statistically significant increase in phosphorylation [p=0.0013, n = 3 independent experiments]. (E–F) Mass rod response recovery (recovery rate plotted to interstimulus time, ERG paired flash paradigm) in bisindolylmaleimide-injected eyes (to 1,25 µM final concentration in the vitreous) versus corresponding control eyes. Two-way ANOVA (mock vs drug injections and IST as factors) with uncorrected Fisher´s test for multiple comparisons, a-wave bisindolylmaleimide vs mock [1200 ms (p=0.0468); 1500 ms (p=0.0263); 2000 ms (p=0.0019), 4 mice analyzed]; and b-wave bisindolylmaleimide vs mock [1200 ms (p=0.0182); 1500 ms (p=0.0374); 2000 ms (p=0.0085), 4 mice analyzed].

## Discussion

In this study we report that retinal IMPDH1 is phosphorylated at up to three residues in vivo: T159/S160 at the Bateman domain; S416 at the mobile flap in the catalytic domain; and S477 at the COOH-terminus. In the bovine system, in which calf eyes were dark-adapted in situ for 1 hr and retinas were processed in darkness or after 5 m of bright light exposure, T159/S160 was preferentially phosphorylated in response to light, S477 in the dark-adapted state, and S416 indistinctly at both conditions (*Figure 2*).

The effect of these phosphorylation events on hIMPDH1 catalytic activity in vitro was determined in enzymatic assays with the corresponding phosphomimetic mutants. The S416D mutation significantly reduces retinal hIMPDH1 catalytic activity. However, because S416 phosphorylation does not appear to be regulated by light, we speculate that this phosphorylation event may respond to metabolic stress or nutrient deficiency (manifested in the phosphoproteomic analysis due to the 1 hr maintenance of the retinas in Locke´s -lacking Ser/Gly- during dark adaptation), to direct IMP towards ATP synthesis. In this respect, S416 maps at a region that resembles a consensus site for AMPK (*Hardie, 2011*), although the AMPK (A2/B2/G1) recombinant human isoform did not recognize hIMPDH1-546aa as a substrate in in vitro phosphorylation assays in this study. We believe that S416 phosphorylation would have a limited contribution to regulation of the enzyme by light in vivo under physiological conditions.

Actually, isoelectrofocusing separation of retinal extracts from dark-adapted or light-adapted living mice revealed that >60% of the protein is phosphorylated under physiological conditions; with the predominant monophosphorylated form corresponding to phosphorylated T159/S160 in the light state and S477 in the dark state (based on the mass spectrometry data); and di-phosphorylated forms likely reflecting slow dephosphorylation kinetics of the reciprocal site.

S477D substitution did not have a direct effect on enzymatic activity neither on GTP allosteric regulation of the enzyme. Thereby, we speculate that S477 phosphorylation might be involved in other aspects of enzyme regulation in vivo: for example inducing filament disassembly. To this respect, a recently reported structure of IMPDH2 polymers has revealed details of the interface that mediates cytoophidia assembly: the 12 amino-terminal residues of the canonical IMPDH2 isoform extend from the catalytic domain to bind into the adjacent molecule, in a shallow surface groove formed by a short helix (476-485), two beta strands (51-63), and two short loops (355-360, 379-380), (*Johnson and Kollman, 2020*). Strikingly, S477 maps into the short helix at the IMPDH2 cytoophidia longitudinal contact interface, allowing us to speculate that phosphorylation of S477 in IMPDH1 might disrupt cytoophidia assembly. Additionally, S477 phosphorylation might also be implied in protein translocation from outer to proximal photoreceptor compartments during dark-adaptation after a period of bright light exposure, or regulating local IMPDH1 interactions with other proteins. Further experiments are needed to corroborate these hypotheses.

We here demonstrate that the T159D and S160D (or T159E and S160E) substitutions increased about 5-fold the $K_{1/2}$ for GDP/GTP inhibitory allosteric regulation of the enzyme in vitro, suggesting that light-dependent phosphorylation of these residues in vivo would effectively desensitize the enzyme to GDP/GTP allosteric control and promote enzyme activation.

Light-dependent activation of IMPDH1 might seem at odds with the assumption in the field that GMP, generated from cGMP hydrolysis during light exposure, would act as a negative feedback regulator of purine nucleotide synthesis by inhibiting IMPDH1, 5-phosphoribosyl 1-pyrophosphate synthetase (PRS), and the first enzyme of the purinosome Glutamine phosphoribosyl amidotransferase (GPA). However, by performing an in vivo metabolic flux analysis of purine nucleotide synthesis in dark/light conditions following an intravitreal injection of labeled glycine, we found that the purinosome activity was substantially increased with light, with IMP levels nearly doubling in light conditions (*Figure 7*). We found that IMP, GMP and AMP levels all increased with light; as did the incorporation of labeled atoms of glycine into these nucleotides. The overall de novo purine nucleotide synthesis increased in the retinas of living mice exposed to bright light.

Considering these results, we believe that most of the GMP increase observed upon light exposure by HPLC actually reflects the light-driven overall increase in de novo purine nucleotide synthesis, rather than the GMP coming from cGMP hydrolysis alone. It is likely that the fraction of GMP coming from the hydrolysis of cGMP would be under detection by HPLC, given that cGMP itself is

under detection by HPLC in retinal extracts. cGMP levels in retinal extracts have been classically determined by radioimmunoassay (*Farber and Lolley, 1974*).

By aligning our results with recent IMPDH structure/function studies (*Buey et al., 2015*; *Buey et al., 2017*; *Fernández-Justel et al., 2019*; *Johnson and Kollman, 2020*), we propose the model of IMPDH1 regulation outlined in *Figure 10*.

In the dark, IMPDH1 would be sensitive to GDP/GTP inhibitory allosteric control (negative feedback in red, *Figure 10*). We have observed that GTP levels in the retina are equimolar or higher than ATP levels (*Figure 5—figure supplement 1*, *Figure 6F*, *Table 2*), as previously reported for frog rod inner/outer segment suspensions (*Biernbaum and Bownds, 1979*) and bovine rod outer segments (*Salceda et al., 1982*). This fact reflects that GTP levels are much higher in photoreceptors than in most cell types, where ATP:GTP levels are between 3:1 to 5:1 (*Traut, 1994*; *Zhao et al., 2015*; *Sumita et al., 2016*).

Upon light exposure, phosphorylation at T159/S160 would dramatically decrease the affinity for GDP/GTP for the canonical site 1 in the Bateman domain, impeding IMPDH1 allosteric inhibition (decreased negative feedback, *Figure 10*). This would be consistent with the increase in the overall flux towards de novo purine nucleotide synthesis measured in light (thicker arrows, *Figure 10*). Furthermore, structural models have shown that depletion of guanine nucleotides induces the assembly of extended octamers into enzymatically active filaments in vitro, that are further stabilized at high IMP levels (*Johnson and Kollman, 2020*). Accordingly, our results suggest that IMPDH1 filamentation is concurrent with the light-driven increase in IMP levels. We speculate that filament formation is what leads to the gradual accumulation of IMPDH1 protein aggregates at the outer segment layer of the retina upon constant light exposure (*Figure 6*). This aggregate formation in vivo is reversible and does not correlate with changes in protein expression, which excludes that it results from transcriptional regulation (*Figure 6*).

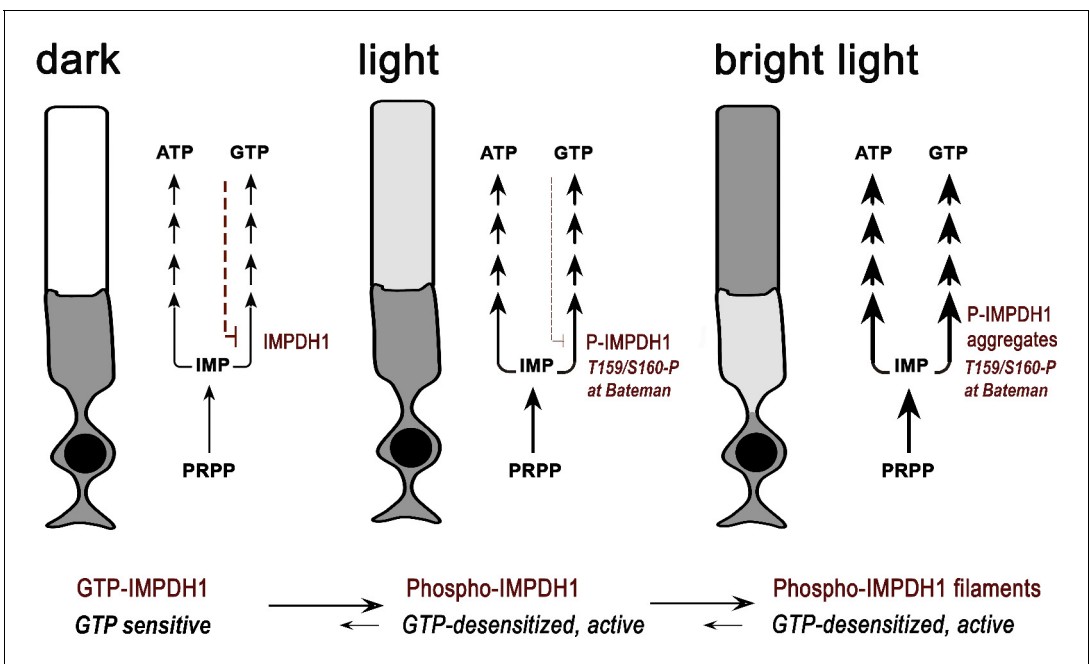

**Figure 10.** Proposed model of regulation of IMPDH1 activity in photoreceptor cells by light-dependent phosphorylation at T159/S160. In the dark, IMPDH1 is sensitive to GTP. GTP can bind to the nucleotide binding sites at the Bateman domain and shift IMPDH1 conformation towards GTP-bound IMPDH1 compacted octamers that are inactive (negative feedback regulation in red). Upon light exposure, phosphorylation at T159/S160 desensitizes the enzyme to GTP, coinciding in time with the increase in IMP levels. We propose that these concurring conditions would cause filamentation of the enzyme, leading to protein aggregation at the outer segment layer. In this way, light exposure would stabilize IMPDH1 in an active conformation permissive to the increased flux towards guanine and adenine nucleotide synthesis (thicker arrows in light). This increased flux is likely required to compensate for ATP and GTP consumption in the phototransduction process, to sustain the pools of ATP and GTP. IMPDH1 localization at the outer segment layer in these macrostructures would be reverted by dephosphorylation and gradual decrease of IMP levels during dark-adaptation. The model would predict a concerted modulation of ADSS/ADSL and the purinosome.

IMPDH cellular aggregates are a landmark of cells or tissues with increased GTP demand and/or circumstances of metabolic GTP deficiency that require stimulation of GTP biosynthesis (*Chang et al., 2015*; *Aughey and Liu, 2016*; *Keppeke et al., 2018*; *Calise et al., 2016*; *Liu, 2016*). A recent in vivo study has shown, for instance, that IMPDH1 formation of filaments is inherent to T cell activation and proliferation, triggered at a post-translational level by NFAT or mTOR signaling, and serves to increase guanine nucleotide levels (*Duong-Ly et al., 2018b*).

IMPDH1 aggregation upon light exposure points to a requirement to expand the GTP pool in the light condition. Furthermore, the increased synthesis of IMP, AMP and GMP points to an expansion of both ATP and GTP pools. In this context, the light exerted regulation on IMPDH1 could serve to promote and stabilize an active conformation of the protein that is permissive to this increased flux to purine nucleotides. Presumably IMPDH1 regulation would be part of a concerted global regulation of the purine synthesis pathway in vivo, involving other metabolic enzymes.

The fact that light-dependent IMPDH1 aggregates accumulate at the outer segment layer suggests that the light induced GTP demand is due to phototransduction. In light conditions, GTP at the outer segment would be consumed in its conversion to GDP by transducin GTPase, and to cGMP by retinal guanylate cyclase activity. By performing simultaneous electroretinogram recordings from both eyes of mice intravitreally injected with an IMPDH inhibitor or control saline buffer we discarded that inhibiting IMPDH had a noticeable effect on activation of the light response (*Figure 8*). However, inhibiting IMPDH caused a delay in mass rod response recovery, that was modest but statistically significant (*Figure 8*). Thereby, we conclude that de novo GTP biosynthesis contributes to cGMP replenishment during the recovery phase of the light response, even if cGMP can be substantially sustained in dark-adapted responses by the *salvage* pathways.

In this respect, retinal GTP levels upon light exposure were reported to be maintained when it was living mice that were exposed to light, or in situ retinas kept in supplemented tissue culture medium (with glycine and serine), like in Du et al. study (*Du et al., 2016*) and this study. In contrast, reports of light causing a clear decrease in GTP come from determinations performed in isolated ROS/RIS preparations or retinas kept in Ringers, that cannot sustain one-carbon metabolism (*Biernbaum and Bownds, 1979*; *Salceda et al., 1982*).

According to our model it might seem surprising that Impdh1-/- mice display only a slowly progressive retinal degeneration. However, despite maintenance of the retinal structure, these mice present gradually diminished ERG responses from normal at 5 m to nearly extinguished at 13 m of age (*Aherne et al., 2004*). Furthermore, compensatory changes like upregulation of IMPDH2 cannot be excluded in this chronic scenario.

We here demonstrate that the light-dependent phosphorylated residues T159 and S160 in retinal hIMPDH1 were selective in vitro substrates for PKCα kinase; but not for the other kinases tested [PKA, PKC, PKG, CamKII, CKII, PKB-Akt and AMPK]. We also show that treatment of retinas with the PKC inhibitor bisindolylmaleimide in situ led to a noticeable decrease of GTP retinal levels upon light exposure (*Figure 9*); and its intravitreal injection caused a delay in mass rod recovery similar to that caused by IMPDH inhibitors (*Figure 9*). We know from our phosphoproteomic analysis that the phosphorylation events that prime PKCα occur in the retina in response to light. Therefore, we propose a model in which light triggers a signaling pathway that activates PKCα, that in turn phosphorylates IMPDH1 at the Bateman domain, desensitizing the enzyme to GTP/GDP inhibition. None of the tested kinases phosphorylated S416 or S477 residues of IMPDH1 selectively, so the signaling pathways involving phosphorylation at these residues remain unknown.

Our proposed model that GDP/GTP allosteric regulation of IMPDH1 is controlled by phosphorylation in vivo to adapt GTP synthesis to the illumination conditions supports our previous hypothesis that IMPDH1 mutations would cause the pathology by resulting in higher than normal IMPDH1 activity; and/or by forming irreversible filaments. Abnormally high IMPDH1 activity could conceivably result in abnormally high cGMP synthesis, which is one of the well-known causes of photoreceptor cell damage (*Wang et al., 2017*); or could cause an ATP/GTP unbalance in darkness. On the other hand, the formation of filaments –in moments of sporadic bright light exposure in the natural world– would be reversible in healthy individuals, but irreversible in adRP10 patients by their impaired capacity to sense GDP/GTP levels. This could result in the gradual accumulation of protein aggregates that would eventually turn toxic.

Most *IMPDH1* mutations in adRP10 and adLCA, including the mutations N198K, R224P and D226N analyzed in this study, desensitize the enzyme to GDP/GTP allosteric inhibition (*Fernández-*

*Justel et al., 2019*). Interestingly, we here show that out of the few IMPDH1 mutations that do not significantly reduce GDP/GTP allosteric inhibition directly, like R105W, V268I and H372P, mutant H372P resulted in abnormally high phosphorylation at T159/S160 by PKCα in vitro (*Figure 9*). This means that the H372P mutation would ultimately result in desensitization of the enzyme to GDP/GTP allosteric inhibition as well, even if H372P mutation would do it indirectly.

This study provides key insights into the complex IMPDH1 regulation in the retina, and proposes that phosphorylation-controlled GDP/GTP allosteric regulation of the enzyme plays an important role in vivo by serving to adjust the rate of de novo GTP synthesis to changes in illumination. We propose that IMPDH1 mutations lead to retinal dystrophies by disrupting this mechanism of control. This study calls for the development of adRP10 and rare adLCA animal models in which to test this model and assay novel therapies.

## Materials and methods

### Ethics statement

Pertaining to animal research, this study was conducted in accordance with the ARVO statement for the use of animals in ophthalmic and vision research and in compliance with acts 5/1995 and 214/1997 for the welfare of experimental animals of the autonomous community (Generalitat) of Catalonia; and approved by the ethics committee on animal experiments of the University of Barcelona (Generalitat Reference #9906, protocols Bell 216/17; 217/17 and 218/17).

### Label-free quantitative proteomic analysis of enriched phosphopeptides from dark- and light-adapted bovine retinas

Fresh calf eyes were obtained immediately postmortem and processed at the slaughterhouse premises. Upon collection, eyes were processed in a dark room by excising the cornea and submerging eyecups in chilled oxygenated Locke's buffer (10 mM Hepes, 20 mM $NaHCO_3$, 112.5 mM NaCl, 3.6 mM KCl, 2.4 mM $MgCl_2$, 1.2 mM $CaCl_2$, 0.1 mM EDTA and 10 mM glucose pH7.4) for 1 hr of dark-adaptation. Retinas were then excised under dim red light. Three retinas were kept for 5 m at room temperature in the dark; while three retinas were exposed to light of 3000 lux for 5 m. Following dark/light exposure retinas were homogenized in homogenization buffer (HB) [20 mM Hepes, 115 mM KCl, 10 mM NaCl, 10 mM $MgCl_2$, 50 mM NaF, 5 mM β-glycerophosphate, 1 mM PMSF and complete protease inhibitor cocktail, pH7.4]. Retinal homogenates were kept at −80°C until further analysis. Subsequently, a basic fractionation protocol was performed to obtain fractions corresponding to soluble, peripheral and membrane proteins from each sample.

Sample preparation for mass spectrometry analysis involved in solution digestion (peripheral and supernatant samples) or Filter Aided Sample Prep (FASP) digestion (*Wiśniewski et al., 2009*) (membrane samples). For in soluble digestion, samples were reduced with dithiothreitol (1 hr, 37°C) and alkylated in the dark with iodoacetamide (30 m, 25°C). Resulting protein extracts were diluted 1/3 with 200 mM NH4HCO3 and digested with LysC (Wako, cat #129–02541) overnight at 37°C; and then diluted 1/2 and digested with trypsin (Promega, cat #V5113) for 8 hr at 37 °C. The peptide mix was acidified with formic acid and desalted with a MacroSpin C18 column (The Nest Group, Inc). For FASP digestion, samples were treated as previously described (*Wiśniewski et al., 2009*). Samples were reduced with dithiothreitol (5 m, 95°C); alkylated in the dark with iodoacetamide (30 m, 25°C) and digested with LysC (Wako, cat # 129–02541) overnight at 37°C and with trypsin (Promega, cat #V5113) for 8 hr at 37 °C, as previously reported. Phosphopeptide enrichment was performed in each sample with the Pierce TiO2 Phosphopeptide Enrichment kit (Thermo Scientific, cat #88301).

Phosphopeptide mixes were analyzed in an Orbitrap Fusion Lumos mass spectrometer (Thermo Scientific, San Jose, CA, USA) coupled to an EasyLC (Thermo Scientific (Proxeon), Odense, Denmark). Peptides were directly loaded onto the analytical column and separated by reversed-phase chromatography using a 50 cm column with an inner diameter of 75 µm, packed with 2 µm C18 particles spectrometer (Thermo Scientific, San Jose, CA, USA). Chromatographic gradients started at 95% buffer A and 5% buffer B with a flow rate of 300 nl/min and gradually increased to 22% buffer B in 79 m and then to 35% buffer B in 11 m. After each analysis, the column was washed for 10 m with 5% buffer A and 95% buffer B. Buffer A: 0.1% formic acid in water. Buffer B: 0.1% formic acid in acetonitrile.

The mass spectrometer was operated in DDA mode and full MS scans with 1 micro scans at resolution of 120.000 were used over a mass range of m/z 350–1500 with detection in the Orbitrap. Auto gain control (AGC) was set to 2E5 and dynamic exclusion to 60 s. In each cycle of DDA analysis, following each survey scan Top Speed ions with charged 2 to 7 above a threshold ion count of 1e4 were selected for fragmentation at normalized collision energy of 28%. Fragment ion spectra produced via high-energy collision dissociation (HCD) were acquired in the Ion Trap, AGC was set to 3e4, isolation window of 1.6 m/z and a maximum injection time of 40 ms was used. All data were acquired with Xcalibur software v3.0.63.

## Data analysis

Proteome Discoverer software suite (v2.2, Thermo Fisher Scientific) and the Mascot search engine (v2.5, Matrix Science; *Bowne et al., 2002*) were used for peptide identification and quantification (*Perkins et al., 1999*). Samples were searched against the Uniprot proteome database corresponding to Bos Taurus (UP000009136), and a list of common contaminants (total entries 24,097), plus all the corresponding decoy entries. Trypsin was chosen as enzyme and a maximum of three miscleavages were allowed. Carbamidomethylation (C) was set as a fixed modification, whereas oxidation (M), phosphorylation (STY) and acetylation (N-terminal) were used as variable modifications. Searches were performed using a peptide tolerance of 7 ppm and a product ion tolerance of 0.5 Da, and identified phosphopeptides were filtered for FDR < 5%. Phosphopeptides were quantified by extraction of their precursor areas using Skyline (version 4.2.0.19009). Light/dark fold change and p values were calculated for phosphopeptides present in two or more replicates of each condition.

## Generation and affinity purification of antibody against bIMPDH1 canonical protein

Anti-IMPDH1 antibody was generated against the bovine canonical recombinant protein (bIMPDH1-514aa). Bovine *IMPDH1* cDNA corresponding to the canonical IMPDH1 (514aa) protein was amplified from total RNA from fresh bovine retinas and cloned into pET15b bacterial expression plasmid (Novagen, Madison, WI, USA) for expression in *E. coli* BL21 (DE3) cells. The expression of recombinant protein was induced in bacterial cultures at $OD_{600}$ = 0.6 for 5 hr at 37°C. Cells were then collected and bacterial inclusion bodies were obtained and solubilized in 6M guanidinium hydrochloride buffer as previously described (*López-Begines et al., 2018*). bIMPDH1-514aa.His was purified by metal chelation using a HiTrap column (GE Healthcare, Chicago, IL, USA) and dialyzed against dialysis buffer (1M urea, 0.4M L-arginine, 20 mM Hepes and 200 mM NaCl).

Purified bIMPDH1(514aa).His was used to immunize two New Zealand White rabbits following a standard 84 day protocol: initial injection of protein (0.5 mg) emulsified in Freund's complete adjuvant, and three boosts of protein (0.25 mg) in incomplete adjuvant at three-week intervals. At exsanguination, blood serum was affinity purified with bIMPDH1-crosslinked Aminolink coupling resin (Thermo Fisher Scientific, Waltham, MA, USA); dialyzed against 0.1M sodium phosphate, 0.15M NaCl, pH 7.2, and concentrated with 10K Amicon devices.

## In situ metabolic labeling

Eight six-week-old C57BL6 mice were dark-adapted overnight. Retinas were dissected under dim red light using a night vision dissecting scope, and placed in individual wells of two 12-well dishes – eight retinas per dish- in 600 µl of Locke's buffer [10 mM Hepes, 20 mM $NaHCO_3$, 112.5 mM NaCl, 3.6 mM KCl, 2.4 mM $MgCl_2$, 1.2 mM $CaCl_2$, 0.1 mM EDTA, 10 mM glucose, sodium succinate, sodium glutamate, vitamin and amino acid supplement, pH7.4] containing 1mCi/ml $^{32}$P-orthophosphoric acid [9.000 Ci/mmol, NEX053010 Perkin Elmer, Waltham, MA, USA]. Retinas were incubated for 90 m at 37°C in a 5% $CO_2$ incubator in the dark, to allow incorporation of the $^{32}$P radionuclide into the ATP pool of retinal cells. One multiwell dish was then placed over the bench at room temperature for 5 m in darkness (dark retinas); and the other was exposed to 2000 lux white light (light retinas) for 5 m. Pools of two retinas were homogenized in HB. Samples were then centrifuged at 14.000 rpm for 20 m, 4°C, to obtain supernatant and pellet fractions. Pellets were resuspended in HB with 1% TritonX100. Immunoprecipitation of IMPDH1 was carried out with 10 µg of anti-IMPDH1 pAb and 50 µl of Dynabeads-protein G (Thermo Scientific, Waltham, MA, USA). Samples were resolved by SDS-PAGE and transferred to nitrocellulose membranes, that were directly exposed to

an X-ray film (AGFA Healthcare NV, Mortsel, Belgium) and subsequently immunoblotted for IMPDH1 [anti-IMPDH1 pAb used as primary Ab; goat-anti Rabbit IREDye 800 (LI-COR, Lincoln, NE, USA) as secondary Ab; and bands visualized using an Odyssey Scan System (LI-COR)]. An immunoprecipitation control was carried with an anti-IgG isotype control.

## Isolectrofocusing separation of IMPDH1

$^{32}$P-labeled immunoprecipitated IMPDH1 from an in situ metabolic labeling procedure as described above was separated by isoelectrofocusing (IEF) to discern the differentially phosphorylated forms of the protein. For that, IMPDH1 immunoprecipitated samples were dissolved in IEF loading buffer [7M urea, 2M thiourea, 4% CHAPs, 40 mM DTT, 2% IPG buffer]. Samples were loaded into prehydrated pH-gradient IEF gel strips (18 cm linear gradient pH3-10 DryStrips from GE Healthcare, Chicago, IL, USA), and run in an Ettan IPGphor3 IEF system (GE Healthcare, Chicago, IL, USA). Gel strips were incubated for 10 m in transfer buffer [25 mM Tris base pH8.3, 190 mM glycine, 0.2% SDS, 20% methanol] and proteins transferred to nitrocellulose membranes by capillary action. Membranes were first exposed to an X-ray film and subsequently immunoblotted for IMPDH1 as indicated above.

For isoelectric focusing analysis of IMPDH1 from retinas of living mice after dark/light adaptation, mice were dark-adapted for 16 hr and exposed to 2000 lux fluorescent light after pupil dilation for 5, 20 or 60 m. Retinas were dissected and homogenized in HB. Retinal homogenates were centrifuged at 13000 rpm for 20 m 4℃ and supernatant fractions were kept. Protein was precipitated with 3 volumes of ice-chilled TCA-acetone buffer (13.3% w/v trichloroacetic acid in acetone, 20 mM DTT), and samples were cooled at −20℃ for 16 hr, and centrifuged at 13200 rpm for 1 hr at 4℃. Protein pellets were washed in acetone, air-dried, resuspended in IEF buffer and resolved by IEF as described.

## Cloning of HsIMPDH1 different spliced forms into pETEV15b and site-directed mutagenesis and bacterial heterologous protein expression

Human IMPDH1 (1–514) ORF was obtained from a cDNA library and cloned into an in house modified pET15b expression vector (*Alonso-García et al., 2009*). A PCR-based strategy was used to insert NH2- and COOH- extra sequences to the ORF of the canonical protein, to generate the hIMPDH1 spliced retinal isoforms (1–546 and 1–595). All DNA constructs were corroborated by DNA sequencing.

Phosphomimetic mutants for enzymatic analysis were obtained by introducing mutations that created individual substitutions T159D, S160D, S416D, S477D into the pETEV15b_HsIMPDH1(546) vector; or in the pETEV15b_HsIMPDH1(514), using the QuickChange II Site-directed mutagenesis Kit (Agilent Technologies).

Phospho-knockout mutants for in vitro phosphorylation assays were obtained by introducing mutations that created triple substitutions that left only one phosphorylation site S160G/S416G/S477G (T159-only); T159G/S416G/S477G (S160-only); T159G/S160G/S477G (S416-only) and T159G/S160G/S416G (S477-only) into the pETEV15b_HsIMPDH1(546). A mutant was also generated with the 4 phosphorylation sites mutated to Gly (4KO mutant).

Blindness associated mutations R105W, N198K, R224P, D226N, V268I and H372P were introduced by site-directed mutagenesis in the pETEV15b_HsIMPDH1(546) vector.

Site-directed mutagenesis was confirmed by sequencing both DNA strands in all generated constructs.

### Purification of IMPDH1 enzymes

Human IMPDH1 enzymes were expressed in *Escherichia coli* strain BL21 (DE3) and purified by nickel-chelating affinity chromatography as described before (*Thomas et al., 2012*). His-tagged proteins were only used for the in vitro phosphorylation assays, while the His-tag was removed from the recombinant proteins for the rest of the assays shown in the manuscript.

## Enzymatic assays

IMPDH activity was studied by monitoring the increase in absorbance at 340 nm upon reduction of NAD$^+$ at 32℃, using 96-well microtiter plates. The reaction buffer was 100 mM Tris-HCl, 100 mM

KCl, 2 mM DTT, pH 8.0. Final enzyme concentrations were set to $100 \mu g m L^{-1}$, $NAD^+$ concentration was fixed at 0.5 mM and IMP concentration varied from 0.04 to 5 mM. Purine nucleotides were assayed at concentrations ranging from 0.09 to 6 mM. Free $Mg^{+2}$ concentration was kept constant at 1 mM, as previously described (*Buey et al., 2017*). The experimental data were non-linearly fitted to the Michaelis-Menten equation of enzyme kinetics using the GraphPad Prism program (GraphPad software) in order to obtain the maximum apparent initial velocity ($V_{max}^{app}$) as a function of the nucleotide concentration. Normalized $V_{max}$ values were calculated by dividing the $V_{max}^{app}$ value in the presence of GTP/GDP nucleotides by $V_{max}$ in the absence of nucleotides.

## In vitro phosphorylation with PKC

For in vitro phosphorylation assays with PKC, 5 µg of recombinant protein HsIMPDH1(546) or its mutants were diluted in 12 µl of dilution buffer [TrisHCl pH 8, 500 mM KCl and 2 mM DTT], mixed with an equal volume of 2x kinase buffer [20 mM TrisHCl pH 7; 20 mM $MgCl_2$; 0.05 mM phorbol 12-myristate 13-acetate (PMA); 0.2 mM $CaCl_2$; 0.2 mM ATP]; 1 µl of $^{32}P$-ATPγ 3000 Ci/mmol (Perkin Elmer, Boston, Masachusetts, USA) and 1 µl of recombinant hPKCα (100 ng/µl, Enzo Life Sciences, Farmingdale, NY, USA). Samples were incubated for 15 m at 30°C, and reactions were stopped by addition of 4x SDS loading buffer. Sample fractions corresponding to 2.5 µg of HsIMPDH1(546) were resolved by 12% SDS-PAGE, and proteins transferred to 0.2 µm nitrocellulose membranes (Bio-rad). The membranes were first exposed to X-ray films (AGFA Healthcare NV, Mortsel, Belgium) for 2 hr; and subsequently immunoblotted for IMPDH1. Membranes were scanned at an Odyssey Scan System (LI-COR) and quantified with the Fiji (ImageJ) software.

## Nucleotide determination in retinal extracts by high pressure liquid chromatography (HPLC)

GMP, GDP, GTP, AMP and ATP levels were analytically measured in perchloric acid murine retinal extracts by ion-pairing high-performance liquid chromatography (HPLC). For in situ experiments, 6–8 week old C57Bl mice were dark-adapted for 16 hr and their retinas were dissected under dim red light. Retinas were placed in DMEM high glucose and either kept in the dark for 5 m at room temperature, or exposed to 2000 lux white light. Immediately after the 5 m dark/light exposure retinas were washed with ice-cold PBS and pools of four retinas were homogenized in 120 µl of 1.2M perchloric acid in PBS. Samples were centrifuged at 16000 rpm for 40 m at 4°C. Deproteinized supernatants were transferred to new tubes, extracted with 1 vol of HPLC-grade chloroform, neutralized, clarified and filtered through 0.22 µm filters (Nanosep MF 0.22 µm, Pall, Westborough, Massachusetts, USA). Samples were kept at −80°C until HPLC analysis. For the in situ experiment to test the effect of PKC inhibition on the dark/light levels of guanine nucleotides, bisindolylmaleimide was added to the medium at 50 nM concentration, and the dark-adapted retinas were incubated in a 5% $CO_2$ incubator for 30 m, protected from light. Retinas were then kept in the dark for 5 m, or exposed to 2000 lux white light for 5 m and processed as above. For the in vivo experiment, retinas were obtained from 16 hr dark-adapted mice or from mice exposed to 2000 lux white light for 4 hr with pupils dilated, and processed the same way.

HPLC analysis was performed as previously described (*Di Pierro et al., 1995*), with minor modifications. A BRISA LC2 C18, 250 × 4.6 mm, 5 µm particle size column (Teknokroma, Sant Cugat del Vallés, Spain) was equilibrated with mobile phase eluent A (10 mM tetrabuthylammonium hydroxide, 10 mM $KH_2PO_4$, 0.25% v:v methanol, adjusted to pH 7.00 with HCl). A linear gradient was formed with eluent B (2.8 mM tetrabuthylammonium hydroxide, 100 mM $KH_2PO_4$, 30% v:v methanol, adjusted to pH 5.5 with HCl) as follows: 5 m 100% eluent A; 5 m 90% eluent A; 5 m 70% eluent A; 10 m 63% eluent A; 5 m 55% eluent A; 10 m 45% eluent A; 20 m 25% eluent A; 20 m 0% eluent A (100% eluent B). A Waters Acquity HPLC with a 210 nm-380nm diode array detector was used, at a flow rate of 0.8 ml/min and constant temperature of 40°C. Injector temperature was 10°C. Standard stock solutions (GMP, GDP, GTP, AMP, ATP) were diluted in 0.1M $KH_2PO_4$. Nucleotide concentrations in each sample were determined by measuring the area of the corresponding peak and comparing it to the peak area of the corresponding standard.

## In vivo metabolic flux analysis by injection of a stable isotope of gly

A group of 5 week-old C57Bl mice was dark-adapted overnight. They were anesthesized with an intraperitoneal injection of ketamine (70 mg/kg; Ketalar, Parke-Davis, Wellington, New Zealand) and xylazine (7 mg/kg; Rompun, Bayer, Leverkusen, Germany) in saline solution (NaCl 0.9%), and injected intravitreally with 1 μl of a 20 mM solution of $^{13}C2; ^{15}N$-Glycine [Cambridge Isotope Laboratories CNLM-1673-H-0,5] in phosphate saline buffer. The injection was performed under dim red light. Mice were kept in darkness, or immediately transferred to a bright light setting (1600 lux white light) for 4 hr. Mouse recovery from anesthesia in the dark or light setting was promoted by keeping the mouse body temperature with a heating pad. After 4 hr, retinas were collected, frozen in dry ice and kept at −80°C until mass spectrometry analysis.

Metabolites were extracted from the retina by vortexing retinas in cold acetonitrile: methanol: water (5:4:1 v:v:v) in 1.5 ml tubes protected from light. After vortex, samples were subjected to four cycles of freezing/thawing by immersion in liquid nitrogen and sonication. Samples were kept in ice for 1 hr and then clarified by centrifugation (15200 rpm, 10 m, 4°C).

Extracts were analyzed by ultra high performance liquid chromatography coupled to a 6490 triple quadrupole mass spectrometer (Agilent Technologies) with electrospray ion source (LC-ESI-QqQ) working in positive mode. 10 μl of extract were injected to the liquid chromatographic system. Metabolites were separated using an InfinityLab Poroshell 120 HILIC-Z, 2.1 × 100 mm, 2.7 μm (PEEK lined) column (Agilent Technologies). Mobile phase A was water with 50 mM ammonium acetate and 5 μM medronic acid, and mobile phase B was acetonitrile. Separation of the samples was conducted under the following gradient: 0–0.5 min 80% of B; 0.5–7.5 min decrease to 70% of B; 7.5–8.5 min decrease to 50% of B; 50% B was maintained for 30 s; 9–9.5 min raise to 80% of B; 9.5–11 min 80% of B. The flow was 0.7 ml/min. The mass spectrometer parameters were: drying and sheath gas temperatures 270°C and 400°C, respectively; source and sheath gas flows 12 and 12 l/min, respectively; nebulizer flow 20 psi; capillary voltage 3000V; nozzle voltage 1500V; and iFunnel HRF and LRF 70 and 80V, respectively. QqQ worked in multiple reaction monitoring (MRM) mode using two transitions (quantitation and confirmation transition) for each compound. The transitions for unlabelled metabolites and the collision energy (CE(V)) were: AMP 348→136(16), 348→119(56); ATP 508→136 (32), 508→410(12); GMP 364→152(8), 364→135(52); GDP 444→152(32), 444→135(60); GTP 524→152(32), 524→135(56); cGMP 346→152(12), 346→135(54); IMP 349→137(16), 349→110(60); Gly 76→30(8),76→58(20). For labelled metabolites were: AMP 351→139(16), 351→122(56); ATP 511→139(32), 511→413(12); GMP 367→155(8), 367→138(52); GDP 447→155(32), 447→138(60); GTP 527→155(32), 527→138(56); cGMP 349→155(12), 349→138(54); IMP 352→140(16), 352→113 (60); Gly 79→32(8), 79→61(20), as stated in *Table 3*. The percentage of labeled metabolites in the total pool of a particular metabolite was calculated as '[labeled/(labeled + unlabeled)]x100'. The average percentage of labeled (*) metabolites in the samples were: % IMP* [dark samples: 0,64%; light samples: 1,7%]; % AMP* [dark samples: 0,31%; light samples: 0,53%] and % of GMP% [dark samples: 0,33%; light samples: 0,52%]. This percentage of incorporation of label from $^{13}C2; ^{15}N$-Glycine into purine nucleotides is in line with the reported range of incorporation [0–0,8%] in other tissues (*Fan et al., 2019*), given that exogenous glycine is much less efficient than endogenous glycine at fueling purine nucleotide synthesis in vivo. However our results were very robust in that the experimental ratio of the isotopologues M+3/M+0 for each metabolite was two orders of magnitude higher in the samples than in the corresponding non-labeled metabolite standard. The abundance of M+3 at metabolite standards reflects the natural abundance of isotopologues or 'background labeling'.

## Electroretinogram analysis

A group of four mice was used for benzamide (BZM) injection, and a group of 7 mice for mycophenolate mofetyl (MMF) injection. A group of four mice was used for injection of bisindolylmaleimide PKC inhibitor. Dark-adapted (>12 hr) animals were anaesthetized with an intraperitoneal injection of ketamine (70 mg/kg; Ketalar, Parke-Davis, Wellington, New Zealand) and xylazine (7 mg/kg; Rompun, Bayer, Leverkusen, Germany) in saline solution (NaCl 0.9%) and pupils were dilated with one drop of 1% tropicamide. For each mouse the right eye was injected with the drug of interest, and the left eye with the physiological saline buffer used for drug dilution. MMF was injected intravitreously to an effective concentration of 200 nM (1 μl injection of 1 μM stock); benzamide riboside

**Table 3.** Multiple reaction monitoring (MRM) transitions in LC-QqQ MS.

| Compound | RT (min) | Quantitation transition (CE) | Confirmation transition (CE) |
|---|---|---|---|
| AMP | 3.21 | 348 → 136 (16) | 348 → 119 (56) |
| AMP[*] | 3.21 | 351 → 139 (16) | 351 → 122 (56) |
| ATP | 5.91 | 508 → 136 (32) | 508 → 410 (12) |
| ATP[*] | 5.91 | 511 → 139 (32) | 511 → 413 (12) |
| GMP | 4.60 | 364 → 152 (8) | 364 → 135 (52) |
| GMP[*] | 4.60 | 367 → 155 (8) | 367 → 138 (52) |
| GDP | 6.13 | 444 → 152 (32) | 444 → 135 (60) |
| GDP[*] | 6.13 | 447 → 155 (32) | 447 → 138 (60) |
| GTP | 7.07 | 524 → 152 (32) | 524 → 135 (56) |
| GTP[*] | 7.07 | 527 → 155 (32) | 527 → 138 (56) |
| cGMP | 1.38 | 346 → 152 (12) | 346 → 135 (54) |
| cGMP[*] | 1.38 | 349 → 155 (12) | 349 → 138 (54) |
| IMP | 3.82 | 349 → 137 (16) | 349 → 110 (60) |
| IMP[*] | 3.82 | 352 → 140 (16) | 352 → 113 (60) |
| Gly | 1.79 | 76 → 30 (8) | 76 → 58 (20) |
| Gly[*] | 1.79 | 79 → 32 (8) | 79 → 61 (20) |

[*]labeled (isotopologue M+3); RT: retention time; CE: collision energy (eV).

was injected to an effective concentration of 80 µM (1 µl injection of 400 µM stock); and bisindolyl-maleimide was injected intravitreously to an effective concentration of 250 nM (1 µl injection of 1,25 µM stock).

Two corneal electrodes were used to record simultaneous ERGs from both eyes (Burian-Allen, Hansen Ophthalmic Development Lab, Coralville, IA). Electrodes were placed in the visual axis 2–3 mm from the cornea, with a drop of 2% methyl-cellulose (Methocel, Ciba Vision, Hetlingen, Switzerland) between corneas and electrodes. Mice were maintained for >10 m in absolute darkness before the recordings, initiated 20 m after drug injection. Mouse temperature during the recording was maintained at 37°C with a heating pad (Hot-Cold, Pelton Shepherd Industries, Stockton, CA). Full-field flash ERG was performed, with the retina illuminated with a Ganzfeld dome. To test the effect of the drug under study on rod mass recovery after a saturating light flash, we applied the paired-flash paradigm (*Lyubarsky and Pugh, 1996*), by triggering a test flash (3 Cd·s/m$^2$) followed by a probe flash of identical intensity at increasing inter-stimulus (IST) intervals: 400, 600, 800, 1200, 1500 and 2000 ms (double flash). The rate of recovery for the a- and b-waves at each IST was obtained by calculating the ratio of a- or b-wave of the probe flash to the a- or b-wave of the test flash. Cone responses were recorded following 5 m of light-adaptation with background white light (50 Cd/m$^2$) to a flash light of 3 Cd·s/m$^2$. Recorded electrophysiological responses were amplified; filtered (CP511 AC amplifier; Grass Instruments, Quincy, MA), and digitalized (ADInstruments Ltd, Oxfordshire, UK). The recording process was controlled with Scope version 3.8.1 software (Power Lab, ADInstruments Ltd). The stimulation protocols were designed according to the International Society for Clinical Electrophysiology of Vision.

## IMPDH1 immunofluorescence localization in retinal sections

For the analysis of spicula formation in response to mycophenolic acid (MPA) treatment, retinas from 16 hr dark-adapted mice were dissected under dim red light and placed in DMEM-high glucose in the presence or absence of 10 µM MPA, protected from light in a 5% CO2 incubator for 1 or 2 hr. Retinas were then fixed and embedded in acrylamide as described (*Hoyo et al., 2014*). Cryosections were obtained at 18 µm thickness in a CM15105 Leica Cryostat (Leica Microsystems). Sections were subjected to an antigen retrieval protocol [2 m incubation with proteinase 0.05 mg/ml proteinase K in PBS pH 7.4 and an 8 s heat shock at 70°C]. Sections were then incubated with blocking solution (3% normal goat serum, 1% BSA, 0.3% Triton X-100 in PBS pH 7.4) for 1 hr at room temperature;

with primary antibodies overnight at 4°C [anti-IMPDH1 pAb and anti-Rhodopsin 1D4 mAb]; with Lectin peanut agglutinin (PNA) conjugated to Alexa Fluor 647 (Thermo Fisher Scientific) for 2 hr at room temperature to label cones; and with secondary antibodies anti-rabbit Alexa-488 and anti-mouse Alexa-555 (Thermo Fisher Scientific) for 2 hr at room temperature. Sections were mounted with Mowiol (Calbiochem).

For analysis of the effect of living mice exposure to constant bright light, 16 hr dark-adapted mice were exposed to 1600 lux bright light for 20 m, 2 hr or 4 hr after pupil dilation with a drop of phenylephrine and tropicamide. Retinas were processed and IMPDH1 immunolocalization assays were performed as indicated above.

For 15d constant-dark or 15d constant-light rearing, 4wk-old mice were either housed in a dark cabinet or in a set of constant bright light exposure (1600 lux) for 15 days. Mice were sacrificed, retinas were processed and IMPDH1 immunolocalization was analyzed.

Confocal microscopy images were acquired with a confocal laser scanning microscope Leica SP5 equipped with two HyD detectors. The spectral bands were set to 491–546 nm and 566–663 nm for 488 nm and 555 nm, respectively. The used objective was a HC PL APO 100x/1.4NA Oil and the samples were mounted on type 1.5 coverglasses (170 mm thick). The stacks covered approximately 2 μm in the z-axis with a 0.13 μm step size. The settings of the image acquisition provided a pixel size of about 74.8 nm x 74.8 nm.

The image analysis of the immunofluorescence data was performed using Fiji (ImageJ). In order to maximize the signal to noise ratio, we used the average projection of groups of 3 planes from large z-stacks. For the particle quantification, we selected a region of interest (e.g. magnified frames of rod outer segment layer) and we applied a light Gaussian filter (radius of one pixel). To determine the size of the areas with concentration of fluorescence we selected the pixels from the image that had a signal beyond a certain value. This value was determined from the statistics of intensity, taking only the 8% of the brighter pixels. Then, the areas of the spots were analyzed by using the plugin 'analyze particles' from Fiji. Counted spots were plotted according to its size and the statistical analysis were performed with Graph Pad Prism (unpaired t-tests).

## Acknowledgements

We are very grateful to Eduard Sabidó and Guadalupe Espadas from the CRG/UPF Proteomics Unit for the technical execution of the retinal phosphoproteome analysis. The CRG/UPF Proteomics Unit is part of the Spanish infrastructure for Omics Technologies (ICTS Omics Tech) and it is a member of the ProteoRed PRB3 consortium which is supported by grant PT17/0019 of the PE I+D+I 2013–2016 from the Instituto de Salud Carlos III (ISCIII) and ERDF. We acknowledge Dr. A Gimeno and L Gómez-Segura at the UB Vivarium facility for assistance with antibody generation in rabbits. We are grateful to L Ramirez for technical support in visual neurophysiology experiments. We are grateful to JC Perales for a critical reading of the manuscript. JA and PL acknowledge financial support from the Spanish Ministry of Economy and Competitiveness through the 'Severo Ochoa' program for Centres of Excellence in R and D (SEV-2015–0522), from Fundació Privada Cellex, Fundación Mig-Puig, from Generalitat de Catalunya through the CERCA program and Laser lab Europe (No. 654148). PdV was supported by a grant from the Spanish ISCIII (PI18/00754). RMB was supported by a grant from the Spanish Ministerio de Ciencia, Innovación y Universidades (BFU2016-79237-P). DFJ was supported by a predoctoral contract from the 'Junta de Castilla y León'. AM acknowledges funding from the Ministerio de Ciencia, Innovación y Universidades (BFU2016-80583-R); and the Ramón Areces Foundation: XVII Edition on Rare Diseases; and JA, PL and AM acknowledge funding from the Foundation La Marató de TV3 (Ref. 20141730).

## Additional information

### Funding

| Funder | Grant reference number | Author |
| --- | --- | --- |
| Ministerio de Economía y Competitividad | BFU2016-80583-R | Ana Méndez |
| Fundación Ramón Areces | XVII Edition Rare Diseases | Ana Méndez |

| | | |
|---|---|---|
| Fundació la Marató de TV3 | 20141730 | Jordi Andilla<br>Pablo Loza-Alvarez<br>Ana Méndez |
| Ministerio de Economía y Competitividad | BFU2016-79237-P | Ruben M Buey |
| Instituto de Salud Carlos III | PI18/00754 | Pedro de la Villa |
| Junta de Castilla y León | Graduate student fellowship | David Fernández-Justel |
| Ministerio de Economía y Competitividad | SEV-2015-0522 | Pablo Loza-Alvarez<br>Jordi Andilla |
| Centres de Recerca de Catalunya | CERCA Institutional Support | Pablo Loza-Alvarez<br>Ana Méndez |
| Fundació Privada Cellex | ICFO Institutional Support | Pablo Loza-Alvarez<br>Jordi Andilla |
| Laser Lab Europe | 654148 | Pablo Loza-Alvarez<br>Jordi Andilla |
| Fundación Mir-Puig | | Jordi Andilla<br>Pablo Loza-Alvarez |

The funders had no role in study design, data collection and interpretation, or the decision to submit the work for publication.

## Author contributions

Anna Plana-Bonamaisó, Conceptualization, Formal analysis, Validation, Investigation, Methodology, Writing - review and editing, sample preparation for phosphoproteomic analysis; data interpretation; in situ metabolic labeling and isoelectrofocusing analysis; expression, purification and enzymatic analysis of IMPDH1 phosphomimetic and phosphoknockout mutants; in vitro phosphorylation assays of phosphoknockout and blindness mutants; data analysis of HPLC nucleotide determinations; data analysis of ERG recordings; generation of IMPDH1 antibody and immunolocalization assays. Contributed to drafting of the manuscript; critical reading of final draft; Santiago López-Begines, Investigation, Methodology, Writing - review and editing, sample preparation for phosphoproteomic analysis; cloning of murine and bovine forms of IMPDH1, established methods for protein expression and purification; critical reading of the manuscript; David Fernández-Justel, Investigation, Methodology, Writing - review and editing, cloned the hIMPDH1 cDNAs in expression vectors, analyzed the effect of phosphomimetic mutants on GDP and GTP allosteric inhibition, critical reading of the manuscript; Alexandra Junza, Investigation, Methodology, performed the nucleotide determinations by LC-MS/MS in the in vivo flux analysis; Ariadna Soler-Tapia, Investigation, significantly contributed to establish isoelectrofocusing protocols for retinal extracts; Jordi Andilla, Investigation, Methodology, assisted with image acquisition at the confocal microscope, and with software applications for quantitation of spicula. Critical reading of the manuscript; Pablo Loza-Alvarez, Methodology, assisted with image acquisition at the confocal microscope, and with software applications for quantitation of spicula; Jose Luis Rosa, Investigation, assisted with the in vitro phosphorylation analysis, by expressing and providing us with different kinases; critical revision of the manuscript; Esther Miralles, Investigation, significantly contributed to establish the procedure for detection of nucleotides by HPLC; Isidre Casals, Methodology, conceived the method for nucleotide separation by HPLC, critical assessment of the manuscript; Oscar Yanes, Conceptualization, Investigation, Methodology, Writing - review and editing, established the methodology for nucleotide determination by LC-MS/MS in the in vivo flux analysis; analyzed the data; provided a critical assessment of the manuscript; Pedro de la Villa, Conceptualization, Investigation, Methodology, Writing - review and editing, performed the electroretinogram recordings; analyzed the data; provided a critical assessment of the manuscript; Ruben M Buey, Conceptualization, Formal analysis, Validation, Investigation, Methodology, Writing - review and editing, assisted us with optimized protein expression/purification methods, cloned the hIMPDH1 cDNAs in expression vectors, conceived and supervised the analysis of the effect of phosphomimetic mutants on GDP and GTP allosteric inhibition, significantly contributed to conceptualization of the model and to writing of the manuscript; Ana Méndez, Conceptualization, Formal analysis, Supervision, Funding acquisition, Investigation, Methodology, Writing - original draft, Writing -

review and editing, conceived the experiments; experimentally contributed to most experiments (fine skill manipulations in the dark, e.g. intravitreal injections, retinal dissections/manipulations under infrared setting); directly supervised the execution of most experiments; analyzed and interpreted the data, conceptualized the model and wrote the manuscript

### Author ORCIDs
Santiago López-Begines http://orcid.org/0000-0001-8809-8919
David Fernández-Justel http://orcid.org/0000-0001-5728-2756
Ruben M Buey https://orcid.org/0000-0003-1263-0221
Ana Méndez https://orcid.org/0000-0001-6393-1644

### Ethics

Animal experimentation: Pertaining to animal research, this study was conducted in accordance with the ARVO statement for the use of animals in ophthalmic and vision research and in compliance with acts 5/1995 and 214/1997 for the welfare of experimental animals of the autonomous community (Generalitat) of Catalonia; and approved by the ethics committee on animal experiments of the University of Barcelona (Generalitat Reference #9906, protocols Bell 216/17; 217/17 and 218/17).

### Decision letter and Author response

Decision letter https://doi.org/10.7554/eLife.56418.sa1
Author response https://doi.org/10.7554/eLife.56418.sa2

## Additional files

### Supplementary files

• Transparent reporting form

### Data availability

All data generated or analysed during this study are included in the manuscript and supporting files.

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
