## [Decision Letter]

**Acceptance summary:**

Photoreceptors in the vertebrate retina rely upon cGMP to gate a cation channel, causing it to be open in the dark. The phototransduction pathway results in the destruction of cGMP, and thus channel closure, resulting in hyperpolarization and a reduction in the release of glutamate. The production of GMP, GDP, and GTP is via the enzyme, IMPDH1. The importance of this enzyme to vision is illustrated by the effects of mutant alleles in humans, where retinitis pigmentosa, or loss of night vision followed by loss of daylight vision, occurs with changes at specific amino acids. The regulation of the activity of this enzyme is thus of interest and has been shown to be at least in part regulated by negative, allosteric feedback by GDP/GTP. The study by Plana-Bonamaiso et al. reports that a PKC-dependent phosphorylation event that is regulated by light relieves the feedback inhibition. This mechanism is part of a regulatory pathway that help to keep the levels of cGMP high during light exposure. The data are novel and are part of an emerging story of the regulation of IMPDH enzymes, which are regulated in several ways, including by formation of larger structures, or cytoophidia, and now phosphorylation, and allostery.

**Decision letter after peer review:**

[Editors’ note: the authors submitted for reconsideration following the decision after peer review. What follows is the decision letter after the first round of review.]

Thank you for submitting your work entitled "Phosphorylation-controlled GDP/GTP allosteric regulation sets IMPDH1 activity in the retina." for consideration by *eLife*. Your article has been reviewed by a Senior Editor, a Reviewing Editor, and three reviewers. The following individuals involved in review of your submission have agreed to reveal their identity: Robert Molday (Reviewer #2).

Our decision has been reached after consultation between the reviewers. Based on these discussions and the individual reviews below, we regret to inform you that your work will not be considered further for publication in *eLife*.

Three experts carefully reviewed the manuscript. While the reviewers found the work interesting, the number of substantive questions raised was such that we feel we must reject it. We hope that the reviewers' comments will be useful to you. We apologize for not being able to deliver better news, and we hope that you will continue to consider *eLife* for future submissions.

The reviewers' comments are appended.

Reviewer #1:

The authors have discovered three phosphorylation sites in retinal IMPDH1: (1) S416 phosphorylation changes catalytic activity directly but is light independent; (2) S477 accumulates in the dark and has no apparent effect on function. (3) T159/S160 changes allosteric regulation by GDP/GTP and increases in the light. Poor quality figures and missing experimental details make it difficult to assess these findings. Assuming these issues are oversights, these findings are interesting and significant new insights into the pathological mechanism of IMPDH1 mutations that cause retinitis pigmentosa. The authors also show that IMPDH forms higher order structures in rod cells in response to light and inhibitors. The authors propose an intriguing model for the regulation of IMPDH activity by phosphorylation: light induced phosphorylation of T159/S160 relieves GDP/GTP inhibition, activating IMPDH1 to replenish the guanine nucleotide pool. Unfortunately, the supporting evidence is weak. The paper is also marred by shifting nomenclature and overstatements.

1) The phosphorylation experiments are not consistent: the mass spec experiments suggest that T159/S160 phosphorylation increases with light exposure, but other experiments show that the overall phosphorylation status does not change. This conundrum should be resolved with additional experiments to monitor the phosphorylation status of individual sites. (Subsection “Dark/light-dependent phosphorylation of IMPDH1 predominant retinal isoform”, the statement "monophosphorylated species predominating" is inaccurate. Figure 3D, E show that the mono and di-phosphorylated forms have the same abundance, ~5.9 – the measurements are not accurate to 4 significant figures).

2) The critical panel B of Figure 2 is illegible and the colors are not explained.

3) The retinal guanine nucleotide levels do not appear to change significantly with light. Perhaps this is because the authors are looking at the entire retina, which contains other kinds of cells in addition to photoreceptor cells. Of course, this also means that the phosphorylated IMPDH1 can be in the other cells.

4) Important details are missing in the allosteric regulation experiments – what exactly is V_max_/V_max_^app^? It is not defined in the text or figure legend.

5) Table 1 appears to report a mixture of different constants – some are Ki determined for competitive inhibitors and some are K_1/2_ were more complex sigmoidal inhibition is observed. Note that the text sometimes uses the term Ki for all, other times uses IC50. The nomenclature should be consistent. The title should reflect that both kinds of measurements are reported, as should the column headings. Which substrates were varied, which were held constant? Are the units really mM? The values of Ki for GMP and XMP are 1-2 orders of magnitude greater than expected. The data are not provided. The authors need to report the concentration of NAD and enzyme in the experiment, as well as the temperature and buffer, in the figure and table legends.

6) The phosphomimetic mutations at T159/S160 do have a significant effect on allosteric inhibition, but "abolish" is too strong a term – it is just a 5-fold increase. How exactly is the IC50 for GTP determined? The sigmoidal dependence on GTP concentration is very steep – what is the value of the Hill coefficient? What equation was used?

7) Why does Figure 8 report nucleotide levels in picomole/retina? Isn't picomole/mg protein the better measure? Also, how much of the nucleotides come from photoreceptor cells as opposed to the other cells in the retina?

8) In vitro kinase assays are of questionable physiological relevance.

9) Is the S477 phosphorylation likely to affect formation of rods and rings?

Reviewer #2:

This manuscript aims to elucidate the mechanisms that regulate IMPDH1, an enzyme that catalyzes the rate-limiting step in de novo synthesis of guanine nucleotides, in the retina. Using a range of proteomic, biochemical, physiological and imaging techniques, they show that IMPDH1 isoform in the retina is regulated by light-dependent phosphorylation of T159/S160 residues in the Bateman domain by PKC. This abolishes the inhibition of IMPDH1 by GDP/GTP enabling GTP levels to be maintained during the light response.

This is an interesting study that provides new insight into the allosteric regulation of the enzyme IMPDH1 in the retina. IMPDH1 plays a central role in maintaining guanine nucleotides, key components of the visual response in photoreceptors. The importance of IMPDH1 is highlighted by the finding that mutations in IMPDH1 are known to cause severe retinal degenerative diseases including retinitis pigmentosa (RP10) and Lebers Congenital Amaurosis. Accordingly, insight into the regulatory mechanisms underlying IMPDH1 activity is important for understanding the visual response and molecular mechanisms underlying retinal degenerative diseases. The studies are well done with sufficient statistical analysis and controls. They incorporate both in vitro and in vivo experiments to examine the role of phosphorylation in controlling the GDP/GTP regulation of IMPDH1 activity and provide support for importance of the GDP/GTP allosteric regulation of IMPDH1 as the principle mode of in modulation of this enzyme as earlier proposed by Buey et al. The experiments and analysis also argue against the competitive inhibition by GMP as a primary mode of regulation of IMPDH1. As part of this study, the authors have also determined the effect of several disease-linked mutations on the PKC catalyzed phosphorylation of IMPDH1 at the T159/S160 sites and showed that two mutants (N198K and R224P) have statistically reduced phosphorylation. Although these studies are incomplete since enzymatic activities were not examined, nonetheless they serve as a basis for a more thorough analysis of how these mutations may affect the control of guanine nucleotide levels and lead to disease.

Essential revisions:

Further discussion on the following points would enhance this manuscript.

It is unclear what the underlying mechanism is for light dependent phosphorylation of IMPDH1 at the T159/S160 residues. The authors nicely show that PKC is involved in the phosphorylation. Is PKC activated directly or indirectly in the light or does IMPDH1 undergo a conformational change in the light to enable phosphorylation by PKC? Discussion or speculation on this point would be useful.

The mechanism appears to be mainly involved in controlling guanine nucleotide levels in rods. Is a similar mechanism proposed for cones?

The authors show that the N198K and R224P disease-causing mutations display a decreased level of phosphorylation (Figure 6C). Do these mutations affect GDP/GTP interaction and/or the enzymatic activity of IMPDH1 or just its phosphorylation by PKC? Further discussion would be useful.

Reviewer #3:

IMPDH1 mutations cause retinal degeneration but its mechanisms and regulation are still poorly understood. Plana-Bonamaiso et al. identified four phosphorylation sites, T159/S160, S416 and S477 and found T159/S160 phosphorylation was light-sensitive. Additionally, they measured the activities of phosphorylation-mimic mutants in response to GTP and GDP and identified the upstream kinase for T159/S160. By using ERGs and immunofluorescence with or without IMPDH1 inhibitor in live mice, they conclude that phosphorylation controls GDP/GTP allosteric regulation of IMPDH activity in response to light.

Essential revisions:

1) Light-regulated phosphorylation of T159/S160 is not solid. The difference between light vs dark in Figure 2 is from bovine retinas. The splicing isoforms of IMPDH1 in bovine retina are still unknown although the phosphorylation sites are conservative for known isoforms. As the authors pointed out in the discussion, they were also under nutritional deficit. Additionally, Figure 3C in comparing phosphorylation in mouse isoform is not statistically significant. It will be important to know whether T159/S160 is the major site of phosphorylation in 603/604 aa IMPDH1. It should be easily done by proteomics or PKC inhibitor.

2) The GMP is significantly increased but GTP/GDP remains unchanged by light. However, the authors proposed that GTP/GDP but not GMP regulates IMPDH1 by light. One argument is that Ki of GMP is much higher (1.94 mM) for hIMPDH1-514 than physiological mouse retina. HIMPDH1-514 is not the major isoform in the human retina. (PMID: 16936083) (546 and 595 are the major ones). It is unknown whether hIMPDH1-514 is phosphorylated in vivo (HIMPDH1-514 is the similar isoform to mouse 514aa is not phosphorylated in Figure 3). Furthermore, Ki of GMP varies in big ranges with different IMPDH1 proteins (PMID: 26558346). The second argument is GTP/GMP ratio (GTP is constant but GMP is increased) is more important than GMP in regulating IMPDH1. Is there data that support this hypothesis?

3) The author proposed that light activates IMPDH activity through phosphorylation. Using PKC inhibitor to block the phosphorylation will give a strong evidence. Will the IMDPH1 inhibitor affect GTP, GMP and IMP levels?

4) In the Discussion section "IMPDH1 mutations would cause the pathology by resulting in higher than normal constitutive IMPDH1 activity… Abnormally high IMPDH1 activity could conceivably result in abnormally high cGMP synthesis.,Consequently, every clue would lead to unregulated, constitutive IMPDH1 activity as the underlying cause of the adRP10 phenotype." How would the author explain the discrepancy by other reports: (1) IMPDH1 human mutants have similar IMPDH activity including D226N (PMID: 1829559); (2) Supplement with GTP improves ERGs from IMPDH1 -/- mice (PMID: 19822744).

---

## [Author Response]

[Editors’ note: the authors resubmitted a revised version of the paper for consideration. What follows is the authors’ response to the first round of review.]

Reviewer #1:[…]1) The phosphorylation experiments are not consistent: the mass spec experiments suggest that T159/S160 phosphorylation increases with light exposure, but other experiments show that the overall phosphorylation status does not change.

The overall phosphorylation status does not change when assayed by in situ metabolic labeling with ^32^P incorporation into endogenous ATP. The protein is substantially phosphorylated both in dark and light conditions. This *is consistent* with mass spectrometry results, that revealed that there is one site that is phosphorylated preferentially in darkness, while another is preferentially phosphorylated in light. Therefore, there is ^32^P incorporation at IMPDH1 both in dark and light conditions.

What we wanted to assess by in situ metabolic labeling is that the phosphorylation events detected by LC-MS/MS were happening to a high extent in vivo, to an extent that they would exert an overall effect on the protein. The in situ metabolic labeling revealed something we could not extract from the mass spectrometry result: that >60% of the protein is phosphorylated at any light condition. The electrofocusing experiments confirmed that two sites are phosphorylated.

This conundrum should be resolved with additional experiments to monitor the phosphorylation status of individual sites.

The phosphorylation status of individual sites has been monitored by LC-MS/MS, on dark/light bovine retinas – biological triplicates – (Figure 1), and the statistical results are provided.

The analysis of phosphopeptides by mass spectrometry typically require large amounts of starting material, because the process entails a step of phosphopeptide enrichment (phosphopeptides are much harder to ionize and therefore harder to detect than peptides by mass spectrometry). We used about 0,75 mg total protein in each homogenate (18 samples) in the phosphoproteomic analysis. Because this corresponded to fractionated tissue, we estimate that we would have needed pools of at least 6 murine retinas per sample (~ 108 retinas in all). That is one of the reasons why we performed the phosphoproteomic analysis with bovine retinas. The cost of the LC-MS/MS analysis was over 10,000€, which is why it is not straight-forward to repeat it now in the murine system, with a number of samples that would allow us to perform robust statistics again.

Previous to the phosphoproteomic analysis we had performed IMPDH1 immunoprecipitation in dark/light bovine samples, followed by LC-MS/MS. We had detected peptide A0JNA3[154-169] LVGIVTSRDIDFLAEK with a phosphate assigned to either Thr159 or Ser160 (50:50 probability) in the light sample; and the same peptide unphosphorylated in the dark sample. However, this simple experiment allows the detection of the phosphopeptide, but not a quantitative comparison between samples.

Another attempt at doing quantitative phosphoproteomic analysis in dark/light bovine retinas by iTRAQ (use of isobaric tags for relative and absolute quantitation in proteomic analysis of different samples in a single experiment) and LC-MS/MS failed to reach the depth of analysis required for IMPDH1 detection. These type of phosphoproteomic analysis are not simple, and are very expensive.

In brief, we have demonstrated by in situ metabolic labeling and isoelectrofocusing experiments that the major isoform of IMPDH1 in the murine retina (603/604aa) is phosphorylated at two sites. We have based the identity of those two sites on the mass spectrometry analysis performed with bovine retinas. This is not uncommon in the field. Rhodopsin phosphorylation sites were originally characterized in the bovine system, and were assumed and later shown to be conserved in the murine system. The same happened with phosducin phosphorylation, and with practically any well-established phosphorylation event in signaling proteins in response to dark/light.

Besides, we are providing a supplementary figure that shows that we detected previously reported, well-established phosphorylation sites in rhodopsin, rhodopsin kinase and phosducin with the proper dark/light-preference in our phosphoproteomic analysis.

(Subsection “Dark/light-dependent phosphorylation of IMPDH1 predominant retinal isoform”, the statement "monophosphorylated species predominating" is inaccurate. Figure 3D, E show that the mono and di-phosphorylated forms have the same abundance, ~5.9 – the measurements are not accurate to 4 significant figures).

The statement “monophosphorylated species predominate” is true and accurate in vivo, at any light condition. It is obvious in Figure 3E (arrows at the left added for clarity). Monophosphorylated species (1P arrow) in retinal extracts obtained from living mice are much more abundant than di-phosphorylated species (2P arrow) in any lane.

In Figure 3D, autoradiograph, the di-phosphorylated form of the protein (2P) has incorporated 2 ^32^P atoms, and that is why it reflects twice its abundance. This is now explained in Results text, and in the figure legend. Arrows have been introduced in the figure, to make this clear.

It is true that IP values are not accurate to 4 significant figures. Corrected.

2) The critical panel B of Figure 2 is illegible and the colors are not explained.

This oversight has been corrected.

3) The retinal guanine nucleotide levels do not appear to change significantly with light. Perhaps this is because the authors are looking at the entire retina, which contains other kinds of cells in addition to photoreceptor cells. Of course, this also means that the phosphorylated IMPDH1 can be in the other cells.

In this new manuscript retinal guanine nucleotide levels are provided from extracts from whole retinas obtained from living mice that were dark- or bright light-adapted, determined by HPLC or by LC-MS/MS. We have consistently observed that GMP increases with light, while GTP levels are maintained. We have also observed that AMP levels increase with light while ATP levels are maintained.

It must be noted that although our nucleotide determinations were done in whole retinal extracts, guanine nucleotide levels largely reflect nucleotide content in photoreceptor cells, as cGMP and GTP levels are reduced to less than 20% their values in retinas that lack the photoreceptor cell layer [rd1 mouse model (Du et al., 2016)]. This is now mentioned in this Results section.

There are numerous studies since the 70s that have determined guanine nucleotides levels in whole retinas, in rd1 mice lacking the photoreceptor cell layer (Farber and Lolley´s work, including their seminal paper in 1974); and in the different retinal layers obtained by tangential sectioning [e.g. Berger et al., (1980)]. It is well established that cGMP and guanine nucleotide levels are much higher in the photoreceptor cell layer than at inner retinal layers.

IMPDH1 is expressed to a much higher extent in the photoreceptor cell layer of the retina than at the inner retina. If the gain at the confocal microscope is adjusted to avoid saturation of the IMPDH1 signal at the photoreceptor cell layer, then the signal at the inner retina is negligible. Representative image shown (we normally cut the image, but it can be appreciated that the signal at the inner retina is not noticeable).

4) Important details are missing in the allosteric regulation experiments – what exactly is V_max_/V_max_^app^? It is not defined in the text or figure legend.

V_max_ are the values derived from the non-linear fitting to a Michaelis-Menten or sigmoidal graph of enzyme velocity vs. substrate equations. V_max,_^app^values refer to the apparent V_max_ in the presence of modulators, such as GTP or GDP. This information has now been clarified in the new manuscript.

5) Table 1 appears to report a mixture of different constants – some are Ki determined for competitive inhibitors and some are K_1/2_ were more complex sigmoidal inhibition is observed. Note that the text sometimes uses the term Ki for all, other times uses IC50. The nomenclature should be consistent. The title should reflect that both kinds of measurements are reported, as should the column headings. Which substrates were varied, which were held constant? Are the units really mM? The values of Ki for GMP and XMP are 1-2 orders of magnitude greater than expected. The data are not provided. The authors need to report the concentration of NAD and enzyme in the experiment, as well as the temperature and buffer, in the figure and table legends.

The reviewer is right. We have thought about this issue and have concluded that our competitive Ki constants might be biased by the use of a relatively large concentration of enzyme (100 microg / mL, i.e. 1.8 microM) for the enzyme kinetic experiments. This should not significantly influence the V_max_ values, determined at high substrate concentration (up to 5 millimolar IMP), but the K_m_ values, which are strongly affected by the low substrate concentrations. The lowest IMP concentrations used for these experiments were 19 microM and the enzyme concentration was 1.8 microM. Thereby, these are not Michaelis-Menten conditions, and the K_m_ values for the lowest IMP experiments might have been not reliably estimated. Consequently, the Ki values might be biased by the wrong K_m_ values, since K_m_ is used to calculate Ki.

Therefore, we repeated the enzyme kinetic experiments using a much lower enzyme concentration (50 nM) and monitored the reaction progress using the NADH fluorescence, which is more sensitive than monitoring NADH absorbance, as we used in our previous experimental setup.

The new K_m_ and competitive Ki,_GMP_ values are now orders of magnitude lower, in accordance to the reviewer´s comment and the previously published literature.

However, the demonstration that de novo synthesis of IMP, GMP and AMP increases with light in vivo, overcomes the discussion on the kinetic constants of inhibition for GMP and GDP/GTP (that has been removed). The ultimate question for our model is whether IMPDH1 activity increases with light in vivo.

According to our HPLC and LC-MS/MS data, light increases the levels of GMP in vivo (as it does increase the levels of AMP and IMP). The overall flux through de novo purine nucleotides increases with light, including IMPDH1 activity.

We have added detailed information about the enzyme kinetics experiments, as requested by the reviewer.

6) The phosphomimetic mutations at T159/S160 do have a significant effect on allosteric inhibition, but "abolish" is too strong a term – it is just a 5-fold increase. How exactly is the IC50 for GTP determined? The sigmoidal dependence on GTP concentration is very steep – what is the value of the Hill coefficient? What equation was used?

The reviewer is right: “abolish” is too strong. We have substituted this term by “decreases” or “desensitizes”. IC_50_ for GTP can only be roughly estimated because higher concentrations of GTP would be needed. This issue has now been clarified in the main text, where we have substituted the “IC_50_ = 6 mM” by “IC_50_ > 6 mM”.

7) Why does Figure 8 report nucleotide levels in picomole/retina? Isn't picomole/mg protein the better measure? Also, how much of the nucleotides come from photoreceptor cells as opposed to the other cells in the retina?

Landmark papers in the field, e.g. [Farber and Lolley, (1974)] have expressed nucleotide levels both ways…it may facilitate the contrast of information among different laboratories. We have now restricted values in Table 2 to those expressed in picomole/mg protein.

*8)* in vitro*kinase assays are of questionable physiological relevance.*

We are now providing new evidence that supports PKC is very likely the kinase responsible for light-dependent IMPDH1 phosphorylation in vivo.

Treatment of retinas in situ with PKC selective inhibitor bisindolylmaleimide results in a drop in GTP levels with light exposure; and treatment of retinas with this drug in living mice causes a delay in mass rod recovery (new Figure9). By determining the kinase, we can now explore the pathway with selective inhibitor drugs.

Results point to light setting in motion a signaling pathway that activates PKC, which impacts main metabolic pathways in photoreceptor cells.

9) Is the S477 phosphorylation likely to affect formation of rods and rings?

Based on the study just published in *eLife* Johnson and Kollman, (2020) we now speculate in the discussion section that S477 phosphorylation might be involved in inducing filament disassembly. In Johnson´s et al., reported structure of IMPDH2 polymers, details are revealed of the interface that mediates cytoophidia assembly: the 12 amino-terminal residues of the canonical IMPDH2 isoform extend from the catalytic domain to bind into the adjacent molecule, in a shallow surface groove formed by a short helix (476-485), two beta strands (51-63), and two short loops (355-360, 379-380) (51). Strikingly, S477 maps into the short helix at the IMPDH2 cytoophidia longitudinal contact interface, allowing us to speculate that phosphorylation of S477 in IMPDH1 might disrupt cytoophidia assembly.

Reviewer #2:[…]Essential revisions:Further discussion on the following points would enhance this manuscript.It is unclear what the underlying mechanism is for light dependent phosphorylation of IMPDH1 at the T159/S160 residues. The authors nicely show that PKC is involved in the phosphorylation. Is PKC activated directly or indirectly in the light or does IMPDH1 undergo a conformational change in the light to enable phosphorylation by PKC? Discussion or speculation on this point would be useful.

This is indeed a very interesting point. T159/S160 residues are exposed to the bulk solvent in the IMPDH1 tetramers, so PKC can easily access them. These residues only get buried when a nucleotide (ATP and or GTP) induces IMPDH octamerization. Thereby, we think it is rather PKC that gets activated in light and phosphorylates IMPDH1.

We have observed in our phosphoproteomic analysis that the phosphorylation sites that prime PKC for activation occur in the retina in the light condition. Thereby, we think it is rather PKC that gets activated in light and phosphorylates IMPDH1.

The mechanism appears to be mainly involved in controlling guanine nucleotide levels in rods. Is a similar mechanism proposed for cones?

This is a very interesting question for which we do not have an answer at this point. Our overall phosphoproteomic analysis reveals that light changes the metabolic fluxes in retinal cells (likely rod photoreceptors) by controlling the phosphorylation of key metabolic switches. We believe that key phosphorylation events rapidly re-structure the metabolic roadmap in photoreceptor cells in dark/light. These metabolic adaptive changes might be different in rods and cones.

The authors show that the N198K and R224P disease-causing mutations display a decreased level of phosphorylation (Figure 6C). Do these mutations affect GDP/GTP interaction and/or the enzymatic activity of IMPDH1 or just its phosphorylation by PKC? Further discussion would be useful.

Both mutations N198K and R224P affect GTP/GDP interaction, but they do not affect enzymatic activity (Fernandez-Justel et al., JMB 2019). N198K is involved in the direct binding of GTP/GDP to the third non-canonical site, which is needed for GTP/GDP-mediated allosteric inhibition. On the other hand, R224P is necessary for GTP/GDP-induced IMPDH octamerization, which is also needed for allosteric inhibition.

We do not know at this point the mechanism by which these mutants display a decrease level of phosphorylation.

Reviewer #3:[…]Essential revisions:1) Light-regulated phosphorylation of T159/S160 is not solid. The difference between light vs dark in Figure 2 is from bovine retinas. The splicing isoforms of IMPDH1 in bovine retina are still unknown although the phosphorylation sites are conservative for known isoforms. As the authors pointed out in the discussion, they were also under nutritional deficit. Additionally, Figure 3C in comparing phosphorylation in mouse isoform is not statistically significant. It will be important to know whether T159/S160 is the major site of phosphorylation in 603/604 aa IMPDH1. It should be easily done by proteomics or PKC inhibitor.

We have demonstrated by in situ metabolic labeling that the major isoform of IMPDH1 in the murine retina (603/604aa) is phosphorylated at two sites. We base the identity of those two sites on the mass spectrometry analysis performed with bovine retinas, that is correct. This is not uncommon in the field. Rhodopsin phosphorylation sites were originally characterized in the bovine system, and are conserved in the murine system. The same happens with phosducin phosphorylation, and with practically any well-established phosphorylation event that happens in signaling proteins in response to dark/light.

Please see the answer provided to reviewer #1 on the sensitivity limitations and the cost of phosphoproteomic experiments.

2) The GMP is significantly increased but GTP/GDP remains unchanged by light. However, the authors proposed that GTP/GDP but not GMP regulates IMPDH1 by light. One argument is that Ki of GMP is much higher (1.94 mM) for hIMPDH1-514 than physiological mouse retina. HIMPDH1-514 is not the major isoform in the human retina. (PMID: 16936083) (546 and 595 are the major ones). It is unknown whether hIMPDH1-514 is phosphorylated in vivo (HIMPDH1-514 is the similar isoform to mouse 514aa is not phosphorylated in Figure 3). Furthermore, Ki of GMP varies in big ranges with different IMPDH1 proteins (PMID: 26558346). The second argument is GTP/GMP ratio (GTP is constant but GMP is increased) is more important than GMP in regulating IMPDH1. Is there data that support this hypothesis?

Regarding GMP Ki values, please see the answer provided to reviewer #1.

Whether GDP/GTP allosteric regulation of IMPDH1 or GMP competitive inhibition of IMPDH1 prevails in vivo… the ultimate question for us is: – *Is IMPDH1 activity increased or decreased with light exposure?* The answer, new Figure 7, is that in vivo the flux through de novo purine nucleotide synthesis increases with light, and that includes IMPDH1 activity (metabolic flux analysis in new Figure 7).

3) The author proposed that light activates IMPDH activity through phosphorylation. Using PKC inhibitor to block the phosphorylation will give a strong evidence. Will the IMDPH1 inhibitor affect GTP, GMP and IMP levels?

Yes, we are now providing a new panel in the PKC figure (Figure 9) that shows just that.

Furthermore, we also show that bisindolylmaleimide injected intravitreally delays mass rod recovery by ERG (paired flash paradigm) in living mice, having a very similar effect to that of inhibiting IMPDH activity.

4) In the Discussion section "IMPDH1 mutations would cause the pathology by resulting in higher than normal constitutive IMPDH1 activity… Abnormally high IMPDH1 activity could conceivably result in abnormally high cGMP synthesis., Consequently, every clue would lead to unregulated, constitutive IMPDH1 activity as the underlying cause of the adRP10 phenotype." How would the author explain the discrepancy by other reports: (1) IMPDH1 human mutants have similar IMPDH activity including D226N (PMID: 1829559); (2) Supplement with GTP improves ERGs from IMPDH1 -/- mice (PMID: 19822744).

1) IMPDH1 human mutants have similar IMPDH1 activity (as the wildtype protein) in in vitro assays, in studies from several laboratories, including ours. Mutations decrease the allosteric inhibition by GDP/GTP. (Fernández-Justel et al., 2019).

2) That a supplement with GTP improves ERGs from IMPDH1-/- mice is exactly what we would expect and is perfectly consistent with our model. (One thing is the autosomal dominant mutations linked to adRP10 retinitis pigmentosa, that we predict would lead to increased cGMP; and another thing is the loss-of-function mouse model, in which the effect would be, precisely, a decrease in GTP levels).